# A Critical Review on the Feasibility of Synthetic Polymers Inclusion in Enhancing the Geotechnical Behavior of Soils

**DOI:** 10.3390/polym14225004

**Published:** 2022-11-18

**Authors:** Abdullah Almajed, Kehinde Lemboye, Arif Ali Baig Moghal

**Affiliations:** 1Department of Civil Engineering, College of Engineering, King Saud University, P.O. Box 800, Riyadh 11421, Saudi Arabia; 2Department of Civil Engineering, National Institute of Technology Warangal, Warangal 506004, India

**Keywords:** clay, geotechnical properties, sand, stabilization, strength, synthetic polymer

## Abstract

Polymers have attracted widespread interest as soil stabilizers and are proposed as an ecologically acceptable means for enhancing the geotechnical properties of soils. They have found profound applications in diverse fields such as the food industry, textile, medicine, agriculture, construction, and many more. Various polymers are proven to increase soil shear strength, improve volume stability, promote water retention, and prevent erosion, at extremely low concentrations within soils through the formation of a polymer membrane around the soil particles upon hydration. The purpose of this work is to provide an overview of existing research on synthetic polymers for soil improvement. A fundamental evaluation of many synthetic polymers used in soil stabilization is provided, Furthermore, the impact of different polymer types on the geotechnical parameters of treated soil was assessed and compared. Limiting factors like polymer durability and the effect of changing climatic conditions on the engineering behavior of the polymer-treated soils have been critically reviewed. The dominant mechanisms responsible for the alteration in the behavior of polymer-soil admixture are reviewed and discussed. This review article will allow practicing engineers to better understand the intrinsic and extrinsic parameters of targeted polymers before employing them in real-field scenarios for better long-term performance

## 1. Introduction

Geotechnical engineers have recently been limited to carrying out construction work on weak soil because of rapid infrastructure development, population growth, and land cost. Thus, the improvement of the mechanical properties of these soils which do not meet the engineering requirement has to be considered before construction activities commences. Soil improvement can alter the engineering properties of soil such as strength, density, liquefaction potential, compressibility, swell and shrinkage behavior, permeability, etc. The various geotechnical technique used for improving the characteristics of the soil includes compaction, drainage methods, vibroflotation, precompression and consolidation, stone columns, grouting and injection, chemical stabilization, soil nailing and reinforcement, geotextiles and geomembranes, thermal methods, construction of moisture barriers, prewetting, or the replacement of unsuitable soils. Chemical stabilization refers to the addition of a chemical admixture to the soil to enhance the engineering properties. This admixture is usually referred to as a soil stabilizer and can be categorized into two major groups: conventional and unconventional. Cement, lime, and fly ash stabilization are the most often used type of conventional stabilization techniques. Cement, lime, fly ash and bituminous material are examples of the conventional chemical stabilizer used by geotechnical engineers [1]. However, the cement manufacturing process, which requires heating limestone at 1450 °C in a kiln, emits an enormous amount of carbon dioxide (CO_2_) into the environment [2]. The process accounts for 8–10% of global anthropogenic emissions and it is estimated to increase in the near future [3,4]. Aside from the issue of significant amounts of greenhouse gas emissions, cement and lime as chemical stabilizer hinders vegetation growth and causes groundwater contamination [5]. As a result of this negative impact, researchers are inclined toward developing an environmentally friendly alternative to improve the properties of weak soil.

Polymers, recycled waste materials such as shredded rubber, crushed glass, or ceramics, salts, fibers, clay additives, electrolyte emulsions, calcium carbonate precipitation, lignosulfonates, and resins are examples of unconventional stabilizers that have been employed. The demand for a comparatively green and sustainable approach to improve soil properties has led to the laboratory simulation of the natural process of soil cementation by inducing carbonate within the soil particles through chemical reactions that occur between reagents. This soil stabilization process can be referred to as a biologically based technique and categorized into two: microbial-induced enzyme precipitation (MICP) and enzyme-induced carbonate precipitation (EICP). These methods have been reported to improve the geotechnical properties of weak soil [6,7,8,9,10,11,12]. MICP is a bio-mediated technology that uses ureolytic bacteria cells to induce carbonate precipitation within porous media. Sporosarcina pasteurii, a non-pathogenic soil bacterium that forms endospores, is the most often utilized urease-producing bacterial strain [13,14,15,16]. Bacterial activity in MICP technology exhibits a well-bonded fabric in soil particles that range from 10–1000 μ and thus, hindered in very fine soils [17]. MICP has been adopted to modify the engineering properties of soils such as the strength, stiffness, and hydraulic conductivity [18,19,20,21,22,23,24,25,26]; stabilize soil against erosion [8,12,27,28,29,30,31], and the remediation of a crack in concrete [32,33]. In a bid to eliminate the timing-consuming procedure required for the cultivation and fixation of ureolytic bacteria in the MICP technique, researchers focused on the use of plant-extracted urease enzyme to induce carbonate precipitation. The precipitated calcite between the soil via the EICP technique has been reported to increase strength, reduces permeability, contains heavy metals, and control erosion [34,35,36,37,38,39,40,41,42,43]. Unfortunately, despite the efficiency of both MICP and EICP in improving the properties of weak soil, there is a major concern with the generation of effluent ammonia ions (NH_4_^+^) which requires addition techniques to remove them from the bio-cemented soil.

Polymers, on the other hand, have gained massive attraction as soil stabilizers and have been suggested to be an environmentally friendly stabilizers for improving the geotechnical properties of soil. Polymers are made up of extremely large molecules that are multiples of smaller chemical components known as monomers. Polymers examined to improve soil engineering characteristics can be divided into two types: natural and synthetic polymers. Natural polymers, also known as biopolymers, are created by the cells of living organisms and are categorized based on the monomers and structure: polynucleotides, polypeptides, and polysaccharides. Synthetic polymers are man-made polymers primarily produced from petroleum products and are divided into three types: thermoplastics, elastomers, and synthetic fibers. The strength of soil treated with a polymer is dependent on the capacity of the polymer to effectively encapsulate the soil particles and the physical qualities of the polymer [44]. Polymer stabilizers have the potential to address the problems of greenhouse gas emissions and groundwater contamination that are associated with traditional methods.

This paper provides an in-depth examination of the existing research on synthetic polymers used in soil improvement. A basic assessment of several synthetic polymers employed in soil stabilization and the dominant reinforcement mechanisms for polymer-soil interactions were presented. Furthermore, an assessment and comparison of the influence of different polymer types on the geotechnical properties of treated soil were performed. Since the durability of the polymer is a major concern, the impact of varying environmental conditions on the engineering properties of the polymer-treated sol was reviewed.

## 2. Synthetic Polymers Application in Geotechnical Engineering

Synthetic polymer is a promising and eco-friendly material that has subsequently gained applications in civil engineering as an alternative soil stabilizer employed either in liquid, powder, or fiber forms. It is otherwise known as man-made polymers which is synthesized by polymerization of chemical molecules. It has been used to improve the properties of soil such as strength, stiffness, permeability, erosion resistance, water stability, volume changes, etc. The most common synthetic polymers used in the geotechnical application are polyacrylamide (PAM), polyethylene (PE), polypropylene (PP), polyurethane (PU), polystyrene (PS) and Styrene Copolymer, polyvinyl acetate (PVA), polyvinyl Alcohol (PVAO), and polyvinylchloride (PVC). Table 1 presents some synthetic polymers and the geotechnical properties of soil it has been used to improve.

### 2.1. Polyacrylamide (PAM)

PAM [-CH_2_CHCONH_2_-] is produced by the polymerization of acrylamide, obtained by the hydration of acrylonitrile [45,46,47]. PAM exists in various forms: anionic, neutral, and cationic charged molecules. The molecular structure of PAM can either be linear or cross-linked, with the linear molecule being water-soluble while the cross-linked molecule is water-absorbent but not water-soluble [48]. PAM is used in the treatment of drinking water, the manufacture of paper, mineral flotation, enhanced oil recovery, stabilizing steep slopes and roadway cuts, strengthening weak soil, improving bearing capacity, and preventing soil erosion [49,50,51,52,53]. PAM is used to stabilize steep slopes and roadway cuts, strengthen weak soil, improve bearing capacity as well as mitigate erosion PAM is the most often utilized synthetic polymer for mitigating construction-related erosion [54]. This polymer is frequently utilized due to its high efficiency, and low cost, and requires a lesser quantity to achieve a desirable result [48]. Cationic-charged PAM molecules are not suitable for erosion control because of their toxicity to aquatic life while anionic PAM is less toxic [55,56].

### 2.2. Polyethylene (PE)

PE is made up of an amorphous crystalline structure that is obtained by the polymerization of ethylene gas [57] and belongs to the group of polyolefin resin family. PE possesses lightweight, insoluble at room temperature, has very low water absorbency, and is extremely chemically resistant. It is principally utilized in the manufacture of plastic bags, films, geomembranes, and containers. The mechanical properties are heavily influenced by various factors such as branching extent and type, crystal structure, and molecular weight [58]. The major classifications of PE are ultra-low-density polyethylene (ULDPE), very low-density polyethylene (VLDPE), linear low-density polyethylene (LLDPE), low-density polyethylene (LDPE), medium-density polyethylene (MDPE), and high-density polyethylene (HDPE) [59]. However, the most common type of PE compound found in geotechnical engineering applications is HDPE.

### 2.3. Polypropylene (PP)

PP is a thermoplastic polymer produced by chain-growth polymerization of rigid, durable, and crystalline propylene monomers. PP appears analogous to PE apart from the pendant methyl (CH_3_) group linked to the main carbon backbone of the polymer structure [60,61]. Because of its low-cost, less toxicity, and biocompatibility, PP is employed in a wide range of applications. It is characterized by high strength, low surface energy, and low gas and liquid permeability [62]. There are three types of PP polymer: homopolymers, homophasic copolymers, and heterophasic copolymers [59]. PP homopolymer has a higher strength-to-weight ratio and is stiffer and stronger than copolymer and thus, is utilized extensively [63]. It is utilized in applications such as packaging, textiles, stationery, laboratory equipment, automotive parts, industrial fibers, food containers, and so on. PP retains most of its mechanical properties at elevated temperatures and is resistive to a wide range of polar liquids, including alcohols, organic acids, esters, and ketones [64]. PP has been used to develop nanocomposites with clay fabrics, which efficiently reduces and absorbs excess water within the clay samples, hence altering the clay plasticity [65].

### 2.4. Polyurethane (PU)

PU polymer is a segmented block copolymer characterized by the presence of a large number of urethane linkages in the backbone [66]. It is synthesized from a chemical reaction between a polyol, a di- or multi-isocyanate, and a chain extender reagent. The most prevalent form of diisocyanate is tolylene diisocyanate [67]. PU forms an expanded elastic solid when it reacts with water [68]. It is employed in the coating industry because it has high mechanical strength, is resistant to chemicals and corrosion, and has great abrasion resistance, toughness, and low-temperature flexibility [69]. The characteristics of PU are heavily influenced by the structure of the polymer backbone [70]. They can be engineered to have high strength or flexibility and toughness for a variety of applications. In general, PU is only soluble in highly polar solvents such as dimethylformamide, dimethylsulphoxide, or N-methyl pyrrolidone. Polyurethane injection systems have been reported to provide outstanding waterproofing performance in civil engineering, tunneling, and underground operations [68]. A summary of a few geotechnical properties of both coarse and fine-grain soils altered by PU is presented in Table 1.

### 2.5. Polystyrene and Styrene Copolymer

Polystyrene (PS) is an aromatic, hydrophobic synthetic polymer obtained from the polymerization of styrene monomer; a liquid hydrocarbon derived from petroleum. This polymer has found application in electronics, medicine, insulation, appliances, automobile, and the food industry. In geotechnical engineering applications, PS is commonly used in either solid, foam, or expanded PS (EPS) forms. EPS is a lightweight polymer with low heat conductivity and moisture absorption. It is made up of around 2–5% PS and 95–98% air [71,72].

Styrene copolymerizes easily with a variety of monomers, including acrylonitrile, butadiene, acrylates, vinyl acetate (VA), vinyl chloride (VC), and so on [73]. The copolymer of styrene with acrylate or butadiene has been considered in geotechnical engineering to improve a certain number of soil properties as presented in Table 1. Styrene Acrylic Emulsion (SAE) polymer is obtained through the polymerization of styrene and other acrylate esters such as butyl acrylate, acrylic acid, etc. It is utilized in the paper, paint, coating, cosmetic, building, and construction industries. SAE polymers have good hydrophobic, heat resistance, adhesion, and aging resistance properties compared with ordinary acrylate emulsions [74].

Styrene-butadiene rubber (SBR) is a synthetic rubber copolymer that may be obtained by polymerizing styrene and butadiene monomers in solution and emulsion [70,75]. The styrene/butadiene ratio affects rubber qualities; the greater the styrene/butadiene ratio, the harder the polymer [76]. SBR is one of the most commonly used polymers in footwear bottoms and heels, tires, conveyors and transmission belts, seals, membranes, hoses, and so on [75,77,78].

### 2.6. Polyvinyl Acetate (PVA)

PVA is a cheap, colorless, nontoxic, biodegradable synthetic resin prepared by the emulsion polymerization of VA [79,80]. PVA is used primarily in textile and paper sizing, adhesives, and emulsion polymerization [81]. The adhesives form transparent, rigid films that are resistant to weather, water, grease, oil, and petroleum fuels [79]. It is commonly used to improve the stress and anti-shrink properties of glass fiber-reinforced plastics [79]. PVA softens when its temperature rises beyond room temperature, and it is less moisture and humidity resistant than thermosetting resins [82]. It is soluble in a wide variety of solvents at room temperature, swells and softens when immersed in water over an extended period, and easily hydrolyzes the polymer to poly(vinyl alcohol) in acids and alkalis [83]. PVA has been used in a wide range of applications in geotechnical engineering as presented in Table 1.

### 2.7. Polyvinyl Alcohol (PVAO)

PVAO is an eco-friendly, nonionic hydrophilic water-soluble, and semicrystalline synthetic polymer formed by alkaline hydrolysis of PVA [84,85,86]. According to Barui (2018) [87], It is insoluble in organic solvents and only slightly soluble in ethanol, and it is the most widely utilized synthetic polymer for biomedical purposes. In addition, it is used in the manufacturing of paper as a binder; in textile warp sizing; in adhesives as an aqueous adhesive solution; and in the creation of other polymers, such as poly(vinyl chloride), as a surfactant for emulsion or suspension polymerization [88]. PVAO has good thermal and mechanical properties due to its better interfacial adhesion with reinforcing materials such as fibers and particles [89]. However, the physicochemical characteristics and specific functional applications are dependent on the degree of polymerization and hydrolysis [90]. Hydrolyzed PVAC has stronger tensile strength and tear resistance than partially hydrolyzed one, which has less crystallinity and hydrogen bonding [83]. PVAO has greatly been considered a soil stabilizer in geotechnical engineering for a wide range of applications as presented in Table 1.

### 2.8. Polyvinyl Chloride (PVC)

PVC is a synthetic polymer obtained by chain polymerization from its monomer, vinyl chloride [58]. It has been widely used in the production of bottles, pipes, cable insulation, floor coverings, plastics, and medical items due to its good mechanical strength, inexpensive cost, and outstanding physicochemical stabilities such as resistance to alkalis, salts, and highly polar solvents [91,92]. PVC degrades in the presence of light at relatively low temperatures of approximately 100 °C to release hydrogen chloride. According to Mckeen (2014) [93], PVC is a flexible or rigid material that is chemically nonreactive, and the rigid PVC is classified into three types: Type I, Type II, and Chlorinated polyvinyl chloride (CPVC). Type II varies from Type I in that it has higher impact values but less chemical resistance. CPVC is more resistant to high temperatures.

**Table 1 polymers-14-05004-t001:** Summary of polymer used to improve geotechnical properties of soil.

Reference	Soil	Test	Factor Considered	Polymer Type
[94]	Silty Clay	Water erosion	Infiltration depth	Aqua-dispersing-nano-binder (ADNB)
			Water stability	
[95]	Clay	Atterberg limit	Polymer content	CBR Plus
		Compaction		
		CBR		
[95,96]		Oedometer		
[96]		UCS		
[97]	Clay	UCS	Polymer content Curing time	Epoxy resin
		Triaxial		
		Split tensile strength		
[98]	Sand	Compaction	Polymer content	
		Ultrasonic pulse velocity	Polymer content	
			Curing time	
			Cement content	
		UCS	Polymer content	
			Curing time	
			Cement content	
[99]	Sand	Erosion	Polymer content	Interpolyelectrolyte Complexes (IPC)
[100]	Sand	Fatigue test	Polymer content	Methylene Diphenyl Diisocyanate (MDI)
		UCS	Polymer content	
			Curing method	
			Curing time	
			Moisture content	
[101,102]	Clay	Free/volumetric swelling ratio	Polymer content	Polyacrylamide (PAM)
[101]		Sorption test		
[102]		Atterberg limits		
		Compaction		
		Oedometer		
[103]		UCS	Polymer content	
			Curing time	
			Curing condition	
		Direct shear	Polymer content	
			Curing time	
[102]		Soil reactivity	Polymer content	
		Cyclic wetting & drying		
		Crack intensity		
[104]	Clayey sand with gravel	Compaction	Polymer content	
		UCS	Polymer content	
			Curing time	
		Hydraulic conductivity	Polymer content	
		Abrasion resistance		
		Water erosion		
		Durability	Capillary rise	
[104]	Sandy clay	Compaction	Polymer content	
		UCS	Polymer content	
			Curing time	
		Hydraulic conductivity	Polymer content	
		Abrasion resistance		
		Water erosion		
		Durability	Capillary rise	
[104]	Silty sand	Compaction	Polymer content	
[104,105]		UCS	Polymer content	
[104,105]			Curing time	
[104]		Hydraulic conductivity	Polymer content	
[104]		Abrasion resistance		
[105]		CBR		
			Curing time	
[104]		Water erosion	Polymer content	
[104,105]		Durability	Capillary rise	
[106]	Sand	Crust thickness & density	Polymer content	
		Moisture retention		
[107]		Durability	Wet-dry cycles	
			Temperature	
			U.V aging	
[106,107]		Penetration resistance	Polymer content	
[106]		Wind erosion		
[108]	Silt	Atterberg limits	Polymer content	
		Compaction		
		UCS	Polymer content	
			Curing time	
			Curing condition	
		Durability	Freezing thaw	
[109]	Clay	Atterberg limits	Polymer content	Polyethylene (PE)
		Compaction		
		CBR		
		Direct shear		
		Oedometer		
[101]	Clay	Swelling test	Polymer content	Polyethylene oxide (PEO)
		Sorption test		
[65,110,111]	Clay	Atterberg limits	Polymer content	Polypropylene (PP)
[65,110,111]		Compaction		
[65,111]		Oedometer	Polymer content	
[110]			Nanocomposite	
			Curing time	
[65,111]		UCS	Polymer content	
[110]			Nanocomposite	
			Curing time	
[65]		Volumetric shrinkage	Polymer content	
[65]		Desiccation cracks		
[111]		Vane shear		
[112]	Clay	Consolidation	Polymer content	Polyurethane (PU)
		UCS		
		Durability (Long & short term)	UV exposure	
			Wet-dry cycles	
			Freeze–thaw cycles	
[113]	Sulfate rich clay	UCS	Polymer content, curing time	
		Free swell		
[114,115,116]	Sand	Direct shear	Polymer content	
[114,115]			Density	
[115,116]			Fiber content	
[114,115,117]		Tensile strength	Polymer content	
[117]			Curing time	
[114,115,117]			Density	
[115,117]			Fiber content	
[114,115,116]		UCS	Polymer content	
[114,115]			Density	
[115,116]			Fiber content	
[118]		Permeability	Polymer content	
			Curing time	
[119]	Clay	Atterberg limit	Polymer content	Polyvinyl acetate (PVA)
		Direct shear		
		UCS		
		Triaxial		
		Oedometer		
[120]		Wind erosion		
[121]		Swell and shrinkage		
[122]	Sandy clay	Atterberg limit	Polymer content	
		Compaction		
		Free swell index		
		UCS		
[123,124]	Sand	Durability	Thermal aging	
[123]			Freeze–thaw cycles	
[124]			Salt	
			Soaking	
			Wet-dry cycles	
[123,124]		UCS	Polymer content	
[124]			Curing time	
[120,123]		Wind erosion	Polymer content	
[120]	Silt	Wind erosion	Polymer content	
[125,126]	Clay	UCS	Polymer content	Polyvinyl alcohol (PVAO)
[125,126]			Curing time	
[125]			Density	
[125,126]		Durability	Polymer content	
[125,126]			Soaking	
[125,126]			Density	
[109]	Clay	Atterberg limits	Polymer content	Polyvinyl chloride (PVC)
		Compaction		
		CBR		
		Direct shear		
		Oedometer		
[95]	Clay	Atterberg limit	Polymer content	Road Packer Plus (RPP)
		Compaction		
		CBR		
		Oedometer		
[127]	Clay	Compaction	Polymer content	SoilTech MKIII
		UCS		
		CBR		
[128]	Sand	Durability	Outdoor exposure	Styrene-acrylic emulsion (SAE)
			Wet-dry cycles	
			Freeze–thaw cycles	
[129]			Soaking	
[128,129]		UCS	Polymer content	
[129]			Curing time	
[128]		Flexural fatigue	Polymer content	
[129]		Permeability		
		Moisture retention	Polymer content	
			Curing time	
[105]	Silty sand	CBR	Polymer content	
			Curing time	
		Durability	Capillary rise	
		UCS	Polymer content Curing time	
[130]	Sand	Direct shear	Polymer content Fiber content	Styrene-butadiene rubber (SBR) emulsion
		Modulus rupture		
		Durability	Temperature	
		UCS	Polymer content	
			Fiber content	
[131]		Compaction	Polymer concentration	
		CBR		
		Direct shear		
[132]	Clay	Cyclic oedometer	Cyclic swell	Urea Formaldehyde Resin/Melamine Urea Formaldehyde Resin (UFR/MUFR)
			Polymer content	
[133]	Peat	UCS	Polymer content	
			Curing time	
		Durability	Soaking	
[134]	Sand	UCS	Polymer content	
[135,136]	Clay	Atterberg limits	Polymer content	Vinyl copolymer
[135,136]		Compaction		
[135]		Oedometer	Polymer content	
			Curing time	
		Hydraulic conductivity		
[135,136]		UCS	Polymer content	
[135,136]			Curing time	
[136]		CBR	Polymer content	
[123]	Sand	Durability	Thermal aging	
[123]			Freeze–thaw cycles	
[129]			Soaking	
[123,129]		UCS	Polymer content	
[129]			Curing time	
[129]		Moisture retention	Polymer content	
[129]			Curing time	
[129]		Permeability	Polymer content	
[123]		Wind erosion	Polymer content	

## 3. Geotechnical Engineering Properties of Polymer Treated Soils

This section discusses previous experiments conducted on polymer-treated soils. The properties of soil have been reported to be altered by the addition of polymers in certain concentrations.

### 3.1. Atterberg Limit

The Atterberg limits test is a characterization test for establishing the critical moisture content at which fine-grained soils changes between different consistency state. They characterize the physicomechanical behavior of soils and are hence critical for civil applications. According to Dolinar and Skrabl (2013) [137], in non-expansive soils, the Atterberg limits are primarily influenced by the size and proportion of clay minerals, whereas in expansive soils, by the quantity of interlayer water, which is primarily related to the type of clay minerals, exchangeable cations, and the chemical composition of the pore water. The Atterberg limit is one of the criteria for identifying and characterizing expansive soils [138].

Figure 1 and Figure 2 shows the influence of different type of polymers at high concentration (i.e., from 0–16%) and low concentration (i.e., from 0–1.25%) on the liquid limit (LL) and plastic limit (PL) of fine-grained soils. Generally, the Atterberg limits of the treated soil exhibited different behaviors based on the polymer types and concentrations as presented in Figure 1 and Figure 2. In some cases, the LL and PL were observed to decrease with an increase in the polymer concentration, while in a few cases, they increased or did not vary significantly. For instance, for the fine-grained soils treated at a high polymer concentration, the addition of PAM to high plastic silt [108] increased the LL as the polymer concentration was increased from 0 to 16% as shown in Figure 1a. In contrast, the addition and increase in the concentration of acrylic polymer (AP) to high plastic silt [139], and PVAO to low plastic silt [140] did not yield any significant impact on the LL as illustrated in Figure 1a. However, an increase in the concentration of PP polymer in high plastic clay [65] and PVAO in black cotton soil [141] showed a significant reduction in the LL. For expansive clay treated with AP [142], the LL was found to decrease with an increase in the polymer concentration. From Figure 1b, the PL of high plastic clay treated with PAM [108], AP [139], and PVC [109] did not achieve significant changes with an increase in the polymer concentration. Though, for silty clay treated with PE [143], expansive clay treated with AP [142] and high plastic clay treated with PVA [122] the PL was reported to increase as the polymer concentration increased. A reduction in the PL was observed with high plastic clay treated with PP [65]. A reduction in the PL was observed with high plastic clay treated with PP [65].

As presented in Figure 2a, for fine-grain soil treated with a low polymer concentration ranging from 0 to 1.5%, the inclusion of road packer plus (RPP) to high plastic clay [144], CBR plus to expansive clay [96], epoxy resin (EP) and PU to dispersive soil [145] resulted in a gradual reduction in the LL as the polymer concentration was increased. However, the addition of PAM to high plastic clay [102] increased the LL as the concentration was increased from 0 to 0.2%, with a further increase in concentration to 0.6% resulting in a slight reduction in the LL. The polymer treatment had a minimal effect on the PL of fine-grain soil except for the PVAO [145] polymer which increased the PL of dispersive soil as the concentration was increased from 0 to 0.5%. Further increase in the concentration by 0.5% resulted in a slight reduction in the PL.

Several reasons have been suggested for the changes in the LL and PL of fine-grained soil with an increase in the polymer concentration. Bekkouche and Boukhatem (2016) [109] attributed the reduction in LL for high plastic clay treated with PVC to the agglomeration of soil particles which therefore provides less surface area and absorbs lower water layers. Azzam (2014) [65] reported that the formation of nanocomposite materials within the clay matrices resulting from the addition of PP polymer significantly decreased both the LL and PL. The PP polymer stabilization improved the net electrical attraction between surrounding clay particles, improving the grain surface against water by producing a hydrophobic composite material. The hydrophilic property of PAM polymer molecules provided increased adsorption sites for water molecules, leading to increased LL and PL [146].

**Figure 1 polymers-14-05004-f001:**
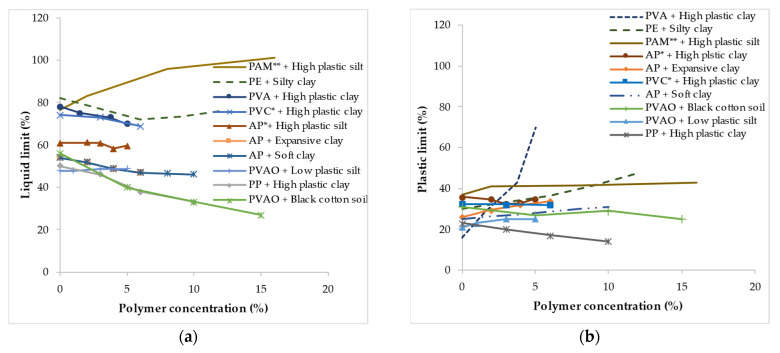
Variation in the Atterberg limit resulting from polymer inclusion at high concentration: (**a**) Liquid limit; (**b**) Plastic limit (* indicates polymer concentration as a % of dry weight of soil and ** as a % of weight of water). Note: these figures were created with data obtained from [65,108,109,122,139,140,141,142,143,147].

**Figure 2 polymers-14-05004-f002:**
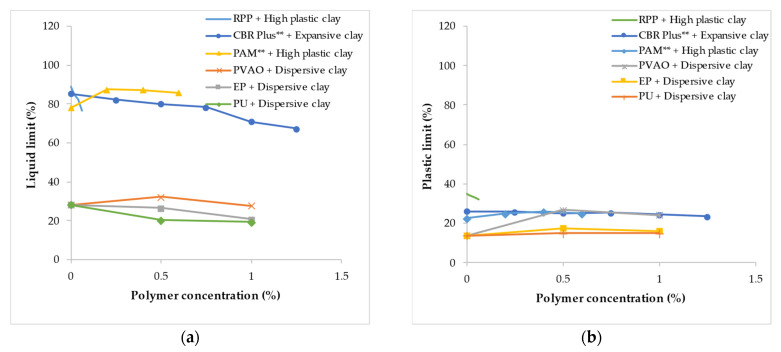
Variation in the Atterberg limit resulting from polymer inclusion at low concentration: (**a**) Liquid limit; (**b**) Plastic limit (** indicates polymer concentration as a % of weight of water). Note: these figures were created with data obtained from [96,102,144,145].

The hydrophilicity of acrylic polymer was reported by Mushtaq and Bhalla (2020) [142] as the factor responsible for the decrease in the LL and increase in PL of expansive clay surrounding clay particles, improving the grain surface against water by producing a hydrophobic composite material. The hydrophilic property of PAM polymer molecules provided increased adsorption sites for water molecules, leading to increased LL and PL [146]. The hydrophilicity of acrylic polymer was reported by Mushtaq and Bhalla (2020) [142] as the factor responsible for the decrease in the LL and increase in PL of expansive clay. In addition, the molar mass of PAM has been reported to have a substantial impact on the liquid limit [148].

Figure 3 presents the location of fine-grained soil treated with polymer on Casagrande’s plasticity chart. From the result, two categories of polymer-treated soil can be observed. One group; high plastic clay [102] and silt [108], treated with PAM illustrated with a dashed oval shows an increase in plasticity index (PI) and increasing LL as the polymer concentration was increased. The other category shows a decrease in plasticity index (PI) and decreasing LL as the polymer concentration was increased. In general, the variation in the PI versus LL follows a linear pattern except for the silty clay treated with PE [143]. For the high plastic clay treated with PAM [102], the mixture with varying polymer concentrations lies within the CH region (high plastic clay). Similarly, high plastic silt treated with PAM polymer [108] lies in the MH region with increasing polymer concentration. However, in soft and expansive clays treated with AP [142,147], an increase in the polymer concentration relocated the soil mixture from CH (high plastic clay) to ML (low plastic silt). The addition of PP [65] to CH relocated the soil to CL as the polymer concentration was increased. The inclusion of PU [145] in dispersive soil shifted the location of the soil on the plasticity chart from CL to CL-ML (silty clay with low plasticity).

### 3.2. Compaction Characteristics

Figure 4 presents the summary of the compaction characteristic conducted on a series of soil treated with different polymer types. The compaction behavior of the soil-polymer mixture varied. For instance, in the case of the two PP polymers used to modify expansive clay. In the first case [65] denoted by ‘PP 1’ in Figure 4, the addition of polymer resulted in a slight increase in the maximum dry density (MDD) as the polymer concentration was increased, and a slight reduction in the optimum moisture content (OMC). However, in the second case [110] represented by ‘PP 2’, the addition of polymer gradually decreased the MDD and OMC as the polymer concentration was increased. It was claimed that the ionic exchange mechanism had an influence on the reduction in OMC, which resulted in the dissipation and absorption of moisture during the chemical process. However, the same reason was suggested to be responsible for both the minor increase and gradual reduction in MDD by Azzam (2014b) [65] and Azzam (2014) [110] respectively. The changes in both cases were attributed to the formation of hydrophobic nanocomposite materials within the soil particles, which acted as nano-fillers. The two studies have argued that the composite material resulted in an increase and reduction in the weight of the soil-polymer mixture, respectively. Bekkouche and Boukhatem (2016) [109] reported a slight increase in the MDD of clayey soil treated with varying concentrations of PVC polymer as shown in Figure 4. However, the OMC was observed to increase significantly following the introduction of 3% of the PVC polymer and a gradual increase was observed as the concentration was increased to 6%. Similar compaction behavior was observed with vinyl copolymer (VC) [136] used to treat clayey soil with high plasticity. The addition of high-density PE to expansive clayey soil slightly reduced both the MDD and OMC as the polymer concentration was increased. The slight changes were ascribed to the reduction of the average unit weight of solids in the soil-polymer mixture. For a sand-bentonite soil mixture treated with acrylamide copolymer (AC) by Ozhan (2019) [149], it is obvious that the MDD and OMC did not vary significantly. The MDD was observed to decrease slightly as the polymer concentration was increased from 0 to 1% and the OMC slightly decreased simultaneously. Beyond a polymer concentration of 1%, the MDD was observed to begin to increase slightly while the OMC decreased slightly. The marginal increase in the MDD was claimed to be a result of the capillary tension of the mixture which resulted in a denser and more stable soil structure. A similar result was reported by the addition of PAM [102] to expansive clayey soil from South Australia.

### 3.3. Strength

#### 3.3.1. Unconfined Compressive Strength (UCS)

The unconfined Compressive Strength (UCS) most commonly used technique for shear strength because it is quick and less expensive to conduct [151]. Figure 5 shows the UCS of polymer-treated fine-grained soil. In general, the addition of polymer to clayey soil did not significantly improve the UCS while in silty soil there was an increase in the UCS with polymer concentration. However, in some instances, a reduction in strength was observed after the optimal polymer concentration was exceeded. Figure 5a shows the UCS of clayey soil treated with different types of polymers. The inclusion of AP1 [147], AP2 [142], and AP3 [152] did not have a significant impact on the UCS. However, the result showed that a further increase in the concentration of AP1 [147] beyond 6% to 8% resulted in a reduction of the UCS by 15% and a further increase to 10% yielded a reduction of approximately 33%. Clayey soil treated with PE [143] showed the highest UCS followed by a commercially available soil stabilizer known as canlite [153]. For the PE polymer [143], the UCS was observed to increase by 108% as the polymer concentration was increased from 0 to 12%. Whereas that of canlite increased by approximately 102% as the polymer concentration was increased from 0 to 15%.

The UCS result of a few polymers used to improve the UCS of silty soil is depicted in Figure 5b. Overall, the highest UCS increment was attained by the Probase polymer [154] while the least was observed with the AP polymer [139]. The inclusion of AP polymer [139] in silty soil only increased the UCS by a magnitude of about 11% as the polymer concentration was increased from 0 to 4% with a further increase to 5% resulting in a 3% loss in UCS. Probase polymer [154] significantly increased the UCS of silty soil by 283% as the concentration was increased from 0 to 8%. A subsequent increment in the concentration by another 8% resulted in a 41% loss in UCS. The loss in UCS was attributed to the fact that the concentration of polymer applied (16%) had exceeded the OMC of the soil-polymer mixture and thus more water molecules are present within the pores which in turn negatively impacts the UCS. Similarly, a reduction in the UCS of silty soil treated with liquid soil stiffer 299 1 (SS299 1) [155] was observed to reach optimal strength at a concentration of 9% which signifies an improvement of approximately 81%. However, a further increase in polymer concentration to 12% resulted in about a 4% loss in UCS. The reduction in UCS was ascribed to an increase in the positive surcharge and the consequent repulsion of soil particles within the soil-polymer mixture, as well as the concentration of alkaline additives that surpassed the requirement for chemical reaction to take place.

Likewise, the UCS of silty soil treated with both canlite 1 [156] and canlite 2 [155], increased by about 117% as the concentration was increased from 0 to 16% and 15%, respectively. The magnitude of UCS developed was less than 5% when the concentration was increased by 8 and 9% to 15 and 16%, respectively. The gain in UCS was suggested to be a result of an electrostatic bond formed between the cationic polymer molecules with the clay particles present in the laterite soil, thus improving the bond between the soil particles [156].

**Figure 5 polymers-14-05004-f005:**
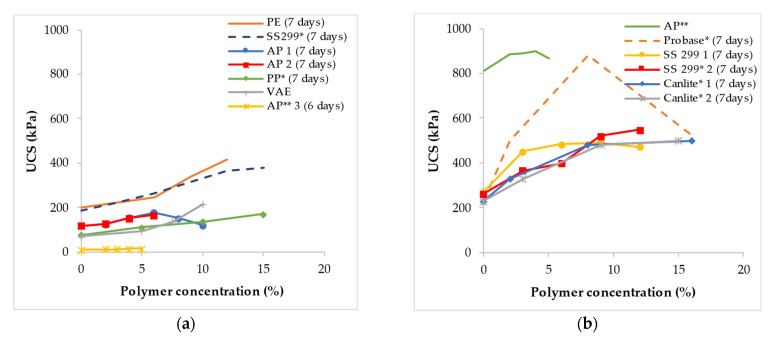
Influence of polymer treatment on the UCS of: (**a**) Clayey soil; (**b**) Silty soil (* indicates polymer concentration as a % of dry weight of soil and ** as a % of weight of water). Note: these figures were created with data obtained from [110,139,142,143,147,152,153,154,156,157,158,159].

The UCS of sand treated with various types of polymers at a high concentration (i.e., from 0–50%) and low concentration (i.e., from 0–4%) is presented in Figure 6a,b, respectively. The polymers used at higher concentrations were reported in the literature to be supplied in the water-based form. The polymers improved the UCS of cohesionless soil by the formation of a polymer membrane within the soil matrix. In general, SAE [128], PVA [124], PMMA [124], and SBR [130] polymers showed higher UCS values compared to the PU polymer [114,116,160,161] as illustrated in both Figure 6a,b. The sand treated with SAE polymer [128] attained a UCS of 5000 kPa at a concentration of 30%, while PU 1–3 polymers [114,160,161] attained UCS values of approximately 281, 169, and 132 kPa, respectively as shown in Figure 6a. For polymers applied to sand at a lower concentration ranging from 0 to 4% as shown in Figure 6b, the UCS achieved at a concentration of 3% were 4380, 3760, and 53 kPa for PVA [124], PMMA [124], and PU [116] respectively. For the SBR emulsion polymer [130], a UCS of 1792 kPa was achieved at a polymer concentration of 2.25%.

Apparently, for most of the polymers reviewed, there was an increase in the UCS with polymer concentration. However, the small growth in UCS of PU-treated sand with polymer concentration was suggested by Liu et al. (2019) [160] to be caused mostly by the unavailability of pore spaces between the sand particles at a particular dry density at which the specimens were prepared. The enhanced interparticle friction and bonding associated with polymer film generated around and in-between soil particles are ascribed to the higher UCS of polymer-treated sand [162]. From the stress–strain relationship, a strain-softening ductile failure behavior has been reported for sand treated with PU [114] and SAE [128] polymers.

In pavement technology, Khoeini et al. (2019) [163] explored the use of a moisture-activated PU liquid polymer, Methylene Diphenylmethane Diisocyanate (MDI), low viscosity, and hydrophobic PU soil polymer grout as a substitute for asphalt binder in wearing surfaces. The UCS of the polymer-limestone aggregate composite was approximately twice that of asphalt. In rammed earth construction application, Yoon et al. (2021) [164] effectively developed a soil stabilization technique based on a water-soluble and environmentally friendly AA and PAM copolymer to improve the renewability and sustainability of soil mixture. The UCS was found to increase with the concentration of AA-co-PAM, regardless of the AA/PAM molar ratio. The UCS increased only until the 7-day curing period, after which the UCS remained constant.

#### 3.3.2. Direct Shear

The direct shear test has also been used to evaluate the efficacy of polymer as a potential soil stabilizer. Figure 7a,b presents the shear strength characteristics) of fine-grained soil treated with various types of polymers. Experimental result has shown that polymer inclusion influences the shear strength characteristics of the fine-grained. For instance, the inclusion of 10% of VA [165] significantly increased the cohesion from 56 to 365 kPa, which indicates an increase of approximately 6.5 folds. However, acetic ethylene ester (AEE) [166] at the same concentration only increased the cohesion by 1.4 folds (from approximately 280 to 397 kPa). The increase in cohesion of specimens treated with AEE polymer [166] was attributed to the low water content resulting from the curing process of the treated soil and rubber friction behavior during the shearing process. Moreover, as the concentration of both VA [165] and acetic AEE [166] polymers increased from 10 to 30%, small growth in cohesion by 5% was observed (i.e., the cohesion increased from 365 to 385 kPa and 397 to 415 kPa, respectively). Furthermore, the cohesion of clayey soil treated with PE and PVC polymer [109] was observed to increase by approximately 10 and 60% as the concentration was increased from 0 to 3%. However, as the polymer concentration was doubled, the cohesion of clay treated with the PVC polymer [109] declined by approximately 18%, while that of high-density PE [109] increased further by approximately 29%. The friction angle of fine-grain soil treated with VA [165] and AEE [166] polymer did not differ significantly as shown in Figure 7b. The friction angle ranged between 29–31° and 56–62°, respectively. The friction angle of fine-grained soil treated with canlite [159] increased by approximately 18% as the polymer concentration was increased from 0 to 9%. Fine-grained soil treated with PVC and high-density PE [109] increased by about 15% as the concentration was increased from 0 to 3%. However, a reduction of about 37 and 17%, respectively was observed as the concentration was doubled.

The shear strength characteristics of cohesionless soil treated with polymer are shown in Figure 8a,b. The cohesion of sand was observed to increase significantly with polymer concentration. The inclusion of 10% of PU1 [114] & PU2 [160] to sand significantly increased the cohesion by 9 and 5 folds, respectively. Further increase in the concentration to 40% yielded an increase in the cohesion by 45 and 20 folds, respectively. For PU3 [116], the cohesion increased from 5.21 to 70.2 kPa as the concentration was increased from 0 to 4%, which signifies an increase of 21 folds. Lastly, for sand treated with SBR emulsion [130], the cohesion was observed to increase from 0 to 69 kPa as the polymer concentration was increased from 0 to 2.25%.

From Figure 8b, the friction angle of sand treated with SBR emulsion [130] was observed to decrease from 36.2 to 32.7, 30.5, 18.5, and 16.9° as the polymer concentration was increased from 0 to 1.50, 1.75, 2.00, and 2.25%, respectively. Likewise, for PU3 [116], the friction angle was observed to gradually decrease from 30.12 to 25.36° as the polymer concentration was increased from 0 to 4%. The inclusion of PU1 [114] in cohesionless soil resulted in an increase in the friction angle from 29.3 to 32.6° as the polymer concentration was increased from 0 to 20%. However, the friction angle was observed to decrease to 24.21° as the polymer concentration was increased beyond 20% to 50%. A similar trend was observed with PU2 [160], where the friction angle was observed to increase from 30.12 to 34.56° as the polymer concentration was increased from 0 to 30%. Further increase in the polymer concentration by 10% resulted in a decrease of approximately 3%. The improvement in the bond between adjacent soil particles and the entwine membrane structure created by the polymer was suggested to be responsible for the increase in the shear strength of the soil [165].

#### 3.3.3. California Bearing Ratio (CBR)

The California Bearing Ratio (CBR) test is a simple test that uses a penetration test on specimens that have been compacted in a laboratory to assess the strength of the material that makes up the base course, subbase, and subgrade of a pavement. The CBR is defined as the ratio of the unit load a circular piston needs to penetrate a test materials with thicknesses of 2.5 mm and 5.1 mm and the unit load needed to penetrate well-graded crushed stone, which is the standard material [167]. CBR values for high-quality sub-base will range between 80% and 100%, while clay typically has a value of 2% and some sands may have values around 10% [168].

The inclusion of polymer to different soil type varied the CBR value as shown in Table 2. Mousavi et al. (2021) [95] utilized CBR Plus and RPP to improve the CBR of clay with high plasticity. The CBR values increased with polymer concentration. Similar observations were reported by Hasan & Shafiqu (2017) [143] and Ahmed and Radhia (2019) [131] following the addition of varying concentration of HDPE and UFR polymer to High plastic clay and sand, respectively. However, in other cases summarized in Table 2, the CBR value increased till an optimum polymer concentration was reached.

The improvement in the CBR may be attributed to the formation of a denser structure as the polymer concentration is increased. The reduction in the volume of soil voids and the effective distribution of soil particles with the polymer particles, according to Bekkouche and Boukhatem (2018) [109], are the reasons why the addition of PVC and HDPE polymer to highly plastic clay results in an increase in CBR.

### 3.4. Hydraulic Conductivity

Figure 9a,b depicts the influence of various polymer type and concentrations on the hydraulic conductivity measurements (seepage of water through the pore voids) for fine-grained and sandy soil specimens respectively. In most cases, there has been a significant reduction in the hydraulic conductivity of soil treated with polymers. However, in a few cases, an increase in hydraulic conductivity has been documented. The addition of 1% of anionic PAM 2a-c [169] polymers to three fine-grained soils as shown in Figure 9a reduced the hydraulic conductivity by approximately 63 to 99% while in the case of PAM1 [170] there was increase in the hydraulic conductivity by about 55% when concentration was increased from 0 to 0.5%. This increase in hydraulic conductivity was associated with the effect of the polymer on soil swelling, which may have disrupted the established surface seal and resulted in a reduction in the bulk density. The effectiveness of PAM on hydraulic conductivity is strongly dependent on the concentration [170]. The inclusion of 5% of PP 1 in expansive clay [171] and 3% of PP 2 in high plastic clay [65] yielded a reduction in hydraulic conductivity by approximately 100 and 68% respectively. A further increase in the polymer concentration to 15 and 10% respectively resulted in more reduction of about 40 and 89%. The modification of microstructures of the high plastic clayey soil caused by induced nanocomposites generated by PP polymer significantly lowered the permeability [65]. Taher et al. (2020) [135] reported that the modification of expansive clay with 5% of VC increased the hydraulic conductivity by approximately 2.5 folds.

On the other hand, from Figure 9b, the addition of 3% of PU1 [118] and PU3 [172] to sand resulted in a significantly reduced hydraulic conductivity by 47 and 84% respectively. A further increase in the polymer concentration to 9% resulted in more reduction of 56 & 85% respectively. The inclusion of 20% of PU2 [160] in sand yielded a decrease of about 67%. Doubling the polymer concentration further reduced the hydraulic conductivity by 82%. This considerable reduction in hydraulic conductivity was attributed to the reduction in pore volume and connectivity caused by the polymeric membranes, which inhibits the flow of water via the limited narrow pore openings [160,172]. According to Al-khanbashi and Abdalla (2006) [162], the modification of sand with 0.5% of three acrylic-based polymers: VA, SA, and AP significantly reduced the hydraulic conductivity by 2, 2, and 5 folds, with signifies a decrease of about 40, 59, and 80% respectively compared to the untreated sand. Further increment in the polymer concentration by a multiple of 10 resulted in more reduction of 96, 93, and 7% respectively. A concentration of 0.5% of these polymers partly covering covered the sand particles by forming a film, which worked as an adhesive that connected the soil particles, resulting in a reduction in hydraulic conductivity. Besides, spraying sand with these polymers rather than mixing yielded more reduction in the hydraulic conductivity [162,173]. For instance, according to Al-khanbashi and Abdalla (2006) [162], spraying the sand with 2% of VA, SA, and AP resulted in 46, 33, and 31% less hydraulic conductivity compared to mixing the sand with these polymers. It was also found that the hydraulic conductivity of sand mixed with these three emulsions polymers reduced as the concentration rose to 3%, and further polymer addition had no significant effect in lowering hydraulic conductivity. Lastly, the hydraulic conductivity of sand mixed with PAM [104] was reported to reduce the hydraulic conductivity by 1.3 times compared to the untreated sand. Andry et al. (2009) [174] and Dehkordi (2018) [175] reported a considerable reduction in saturated hydraulic conductivity for sandy soil modified with different concentrations of hydrophilic isopropylacrylamide polymer and tripolymer of acrylamide, acrylic acid, and acrylate potassium respectively.

### 3.5. Sediment Volume Behavior

#### 3.5.1. Volumetric Swell Ratio (VSR), Free Swell Ratio (FSR) and Free Swell Index (FSI)

Currently, there are limited studies centered on the sediment volume behavior of expansive clay treated with polymers. Inyang et al. (2007) [101] studied the effect of two different molecular weights (high and low) of positively charges PAM and neutrally charged POE polymer on the swell characteristics of Na-montmorillonite. The swell obtained upon the hydration of the soil was termed the volumetric swelling ratio (VSR), defined as the volume of clay at any time relative to the initial volume of clay being immersion. Figure 10 presents the VSR of Na-montmorillonite clay treated with PAM and POE. The polymer with a high molecular weight is denoted by H, while that with a low molecular weight is by L. From Figure 10, both the molecular weight of PAM and high molecular weight POE yielded a VSR that is 3–4 times lesser than that of distilled water as the polymer concentration was increased.

The free swell ratio (FSR) and the free swell index (FSI) have also been used to characterize the degree of expansivity of clayey soil treated with polymers. According to Sridharan and Prakash (2000) [138] and Prakash and Sridharan (2004) [176], the FSR is defined as the ratio of the sediment volume of soil in distilled water to that in carbon tetrachloride or kerosene while the FSI is the percentage of the ratio of the difference between the sediment volume of soil in distilled water and kerosene to that of kerosene. However, in the place of distilled water, the polymer solution was used for the treated clayey soil. Soltani et al. (2018) [102] reported a significant reduction in the FSR of highly expansive clay as the concentration of PAM polymer was increased from 0 to 0.2 g/L, with further increase in the concentration beyond 0.2 g/L leading to a marginal decrease in the FSR as shown in Figure 10b. Based on the classification criteria for expansive soil proposed by Sridharan and Prakash (2000) [138], the inclusion of the PAM polymer improved the high degree of soil expansivity to a moderate level. Also, from Figure 10b, a rapid reduction in the FSI value of expansive clay soil has been reported by Zumrawi and Mohammed (2019) [122] with the addition of PVA. The FSI reduced significantly by 3.8 folds as the PVA polymer concentration increased from 0 to 5%, which signifies a reduction of about 74%. The reduction was ascribed to the polymer physiochemical bonding with the soil particles, which inhibited soil expansivity.

#### 3.5.2. Swell Characteristics (Swell Potential and Pressure)

Expansive clays are one of the most troublesome soils found all over the world, particularly in dry and semi-arid environments. When exposed to variations in moisture content, it suffers considerable volume changes. The presence of highly active clay minerals which can absorb water molecules causes the swelling of expansive soil in general. Thus, an assessment of the swell potential and pressure of expansive soil is critical before the construction of foundations. The feasibility of using polymers to improve the swell characteristics of expansive clay has been reported in the literature.

Figure 11 illustrates the influence of polymer on the swell potential of expansive soil. Azzam (2014) [110] and Azzam (2012) [171] reported a significant reduction in the swell potential by about 79 and 65% respectively following an increase in PP polymer concentration from 0 to 15%. According to the classification procedures for expansive soils established by Sridharan and Prakash (2000) [138] presented in Table 3, the degree of expansivity was treated from ‘very high’ to ‘medium’. For a silty clay-bentonite soil mixture treated with PE polymer, a remarkable reduction in the swell potential by approximately 67% (from 18 to 2.28%) was reported by Hasan and Shafiqu (2017) [143] as the polymer concentration was increased from 0 to 6%. The degree of expansivity of the soil ultimately changed from high to medium. Further increase in the polymer concentration from 6 to 12% resulted in a further decrease in the swell potential and the degree of expansivity became low.

Mirzababaei et al. (2009) [177] evaluated the effectiveness of PMMA and PVA on the swell potential of high plastic clay and silt. The inclusion of a low concentration of PVA to high plastic clay and PMMA to high plastic silt increased the swell potential slightly. A further increase in the concentration generally yielded an insignificant reduction in the swell potential. The PMMA polymer reduced the swell potential of high plastic clay as the polymer concentration was increased from 0 to 5%. As a result, the degree of expansivity changed from high to medium. However, high plastic silt treated with PVA did not have any positive impact on the swell potential and the degree of expansivity was observed to change from medium to high as the polymer concentration was increased from 0 to 5%. The PMMA and PVA polymers were found to be less efficient because the polymer particles were unable to coat the soil particles with a layer of a polymer film. Thus, increasing the free contact boundaries of soil particles and absorbed water. However, when the soil-polymer admixture was allowed to cure under direct sunlight for about two days to lose approximately 90% of its water content, the polymer films coat the soil particles and create a thin film that prevented contact of water molecules with the surface of soil particles and ultimately reduced the swell potential.

In other studies, a remarkable reduction in the swell potential of expansive soil treated with PU [112], RPP [144], VC [135], CBR plus [96], and PAM [102] has been reported. 10% of PU [112] was reported to reduce the swell potential by 90%, 0.06% of RPP [144] by 51%, 5% of VC [135] by 70%, 1.25% of CBR plus [96] by 66%, and 0.2% of PAM [102] by 33%. Soltani et al. (2017) [96] recommended 1% of CBR polymer as the optimum concentration for mitigating swelling of the expansive soil considered since an excessive concentration of 1.25% did not further impact the reduction in swelling significantly. The presence of the CBR plus polymer in the soil was said to improve clay particle hydrophobic behavior by promoting the removal of adsorbed water and preventing water reabsorption, thus reducing the tendency of clay particles to expand.

Significant reductions in the swell pressure of expansive soil treated with polymer have been reported. As shown in Figure 12, the swell pressure of expansive soil treated with PP [110,171], and PE [143] decreased considerably with polymer concentration. The reduction in swell pressure by the PP polymer was attributed to the hydrophobic nanocomposite formed with the clay particles [110].

#### 3.5.3. Compression Index

The compression index, C_c_, is one of the crucial parameters of fine-grained soils used to predict the settlement due to primary consolidation. It is an alternative measure of the compressibility of the soil. There is limited study focused on the influence of synthetic polymers on the compression index of fine-grain soil. The C_c_ of soil treated with polymer has been observed to vary. As shown in Table 4, the inclusion of PVC [109] and HDPE [109] polymer to highly plastic clay yielded a reduction in the C_c_ as the polymer concentration was increased from 0 to 3%. Further increase in the polymer concentration resulted in an increase in C_c_. The addition of 5% of PP [111] polymer to low plastic clay resulted in a reduction in the C_c_ by around 37%. Lastly, the inclusion of Poly(methyl methacrylate-co-butyl acrylate) [MBA] [119], poly(styrene-co-butylacrylate) [STBA] [119], and polyvinyl acetate (PVA) [119] to kaolin resulted in a decrease in C_c_. However, a prominent reduction was observed with the STBA polymer.

## 4. Durability of Polymer Stabilized Soil

The durability and degradability of polymer-treated soil under diverse environmental conditions have been identified as relevant areas of research. From the literature, several parameters have been used to study the durability of polymer-treated soil exposed to extreme environmental conditions. These parameters include and are not limited to; ultraviolet radiation exposure, freeze-thaw cycles, wet-dry cycles, water stability, water immersion, erosion, thermal aging, long-term stability, etc. The efficiency of the polymer with respect to these parameters has been measured by either conducting UCS, direct shear, erosion rate, penetration resistance, or oedometer test.

### 4.1. Freeze-Thaw Cycles

Freeze-thaw (FT) occurs when water present in the pore space of soil freezes, causing the volume of water to expand, thus exerting pressure on the soil particles which may be disruptive and eventually exceed the cohesive bond between adjacent soil particles. The frequent occurrence of the FT cycle induces an alternation of the mechanical stresses exerted when water freezes in pores.

Song et al. (2019) [165] conducted an FT cycle study on clay soil using the optimal VA polymer concentration of 20% obtained from the UCS test. The treated soil was subjected to freezing at −20 °C for 12 h and allowed to thaw at a temperature of roughly 20 °C for 12 h. It was observed that as the number of freeze-thaw cycles increases, the UCS of the treated clay soil gradually decreases. The UCS of the treated soil decreased by approximately 25% (i.e., from 2.36 to 1.78 MPa) following exposure to 10 FT cycles. The reduction in the UCS with an increase in the FT cycle was attributed to the hydration of the polymer membrane on the surface of the soil particles, which caused the phenomenon of frost expansion and weakened the polymer membrane structure. Soltani-Jigheh et al. (2019) [108] reported the influence of varying concentrations of water-soluble cationic PAM on the FT behavior of high plastic silt. As the FT cycles were increased from 0 to 5, the UCS of the specimens was observed to reduce by about 91- 94% across all the polymer concentrations considered. The decline in the UCS was linked to the hydrophilic nature of the PAM polymer and an increase in the water absorption potential of the treated soil with an increase in FT cycles. A similar report was made by Rezaeimalek et al. (2017) [128] with sand treated with SAE polymer and subjected to 24 cycles of freezing at a temperature of −18 °C for 24 h and then thawing in a laboratory environment at 20 ± 2 °C for another 24 h. The UCS was found to have decreased from 5096 to 3051 kPa, demonstrating a 40% drop compared to the unweathered sand specimens, while a maximum weight loss of 9% was noted. Gong et al. (2016) [123] reported insignificant changes in the UCS of sand specimens treated with PVA and VAEC which were subjected to 10 FT cycles of freeze at −18 °C for 22 h and thawed at 25 °C for 2 h. Similarly, Arasan et al. (2015) [178] reported insignificant changes in the UCS of sand treated with polyester (PET) after subjection to a series of 20 FT cycles. The negligible changes in UCS were attributed to the resistance of the polymer to extreme environmental impacts.

### 4.2. Photodegradation

UV radiation induces photooxidative degradation, which results in the breakdown of polymer chains, the production of radicals, and the reduction of molecular weight, resulting in the loss of mechanical characteristics and the formation of worthless materials after an undetermined period [179]. The influence of ultraviolet (UV) and sunlight exposure has been assessed on polymer-modified soil. Song et al. (2019) [165] reported an overall increase in the UCS of clay soil treated with VA polymer exposed to UV illumination for 720 h at a temperature of around 40 °C. The UCS was observed to increase in the early stage of UV exposure from 2.37 to 5.05 MPa at an illumination time of 360 h. However, as the illumination time increased beyond 360 h, the UCS was reported to decrease marginally. The initial enhancement in the UCS at an exposure time of 360 h was attributed to the loss of moisture from the polymer-soil admixture. However, the reduction in UCS was attributed to the prolonged exposure leading to the shrinkage of the polymer-soil mixture beyond tolerable limits. Mirzababaei et al. (2009) [177] reported the influence of UV exposure on the swell potential of three expansive soils treated with two different polymers after 2 days. For all the soil considered, a reduction in the swell potential was reported at a higher polymer concentration of 5% except for high plastic soil treated with PMMA which led to an increase in the swell potential. Rezaeimalek et al. (2017) [128] reported the influence of UV exposure on sand specimens treated with SAE after 8 months. The specimens were reported to appear in a lighter shade in color compared with the unweathered specimens. The UCS of the specimens was reported to slightly reduce from 5096 to 4746 kP after exposure to sunlight for 8 months. The specimen retained approximately 93% of the UCS, which indicated that the UV exposure did not have a significant impact. Ding et al. (2019) [107] reported an insignificant impact on the penetration resistance of soil crust treated with a combination of PAM and xanthan or guar gum of varying mix ratios.

### 4.3. Thermal Degradation

The impact of thermal aging on polymer-treated soil has been considered an area of research when the issue of durability arises. Studies have shown that the exposure of polymer-treated soil to thermal aging has resulted in an increase or decrease in the UCS, depending on the exposure temperature. Gong et al. (2016) [123] reported that exposure of sand specimens treated with PVA and VAEC polymer emulsion to a thermal level of 60 °C for 10 days had no significant impact on the UCS. Zandieh et al. (2010) [124] reported the influence of varying the curing temperature to 22, 40, and 70 °C, on sand specimens treated with a concentration of 1.7% of PVA and PMMA polymer after 24 h. The UCS was reported to increase by 2.3 and 3.1 times as the curing temperature was increased from 22 to 70 °C for PVA and PMMA respectively. In addition, Almajed et al. (2021) [130] reported a decrease of approximately 60% for desert sand treated with 2% of SBR emulsion polymer after subjection to a temperature of 80 °C compared to the unweathered specimen. However, allowing the specimen to cool before testing resulted in some gain in UCS. The UCS was found to be approximately 30% more compared to those tested immediately after thermal curing, but lesser by about 48% compared to the unweathered specimens. The reduction in the UCS of polymer-treated sand was ascribed to the exposure temperature being higher than the heat deflection temperature of the polymer, which lead to the softening of the polymer.

### 4.4. Wet-Dry Cycles

The resistance of polymer-treated soil subjected to alternate wetting and drying is also considered a crucial factor when the issue of durability arises. Rezaeimalek et al. (2017) [128] reported a loss of approximately 39% in the UCS and 7% in weight for sand treated with a concentration of 30% SAE polymer after subjection to 24 wet-dry cycles (24 h of wetting and 48 h of drying). Also, Soltani et al. (2018) [102] reported a reduction in the swell potential of Australian expansive soil treated with 0.2 g/L of PAM polymer and subjected to 5 cycles of wetting and drying. It was suggested that reconstitution of the PAM-clay microstructure after the first or second cycle is responsible for the decrease in swelling potential.

### 4.5. Water Stability and Soaking Test

To further understand the interaction of the polymer-soil admixture with water molecules, a series of water stability and soaking tests with various polymer types and concentrations have been reported. The water stability and the soaking test have been developed to assess the potential of a polymer-soil admixture to withstand deterioration when exposed to water. The water stability index, ‘k’ defines the ability of the treated specimen to resist collapse with respect to time. If the soil collapses after 1 min of immersion, its coefficient is defined as 5, if it collapses between 1–2 min, its coefficient is defined as 15, and if it does not collapse for the whole 10-min immersion, its coefficient is defined as 100 [94,166]. A significant increase in water stability has been reported with an increase in the concentration of ADNB [94], AEE [166], and a crosslinked polymer of PAM and CMC [180]. The stability number increase to 100 at higher polymer concentration, which signifies that the treated specimens have better resistance when exposed to water. The interaction between polymer and soil particles significantly altered the soil fabric, resulting in improved water stability. Qi et al. (2020) [181] reported the water stability of PU-sand admixture based on the disintegration area and shear strength of the submerged specimen. As the PU polymer concentration was increased, the disintegration of the specimen diminished, and the water stability index increased. Significant improvement in the water stability index was reported as the concentration was increased from 0.2 to 0.3%. Beyond this concentration, marginal increment in the water stability was reported as the concentration was increased to 5%.

In other studies, the effect of water on the strength of the polymer-soil admixture has been reported. Liu et al. (2018) [182] reported the influence of immersing PU-treated sand specimens in water on the shear strength parameters. When specimens are submerged in water, the membrane generated by the PU polymer was suggested to soften, potentially changing the behaviors of the admixture. The shear strength of sand treated with 2% PU was reported to decrease with an increase in the immersion time. The cohesion decreased from 53.65 to 26.48 kPa as the immersion time was extended from 1 to 216 h. At a constant immersion time, the shear strength was reported to increase with polymer concentration. In general, the polymer did not have any significant impact on the internal frictional angle irrespective of the concentration or immersion time. Zandieh et al. (2010) [124] reported a reduction of around 57–70% in the UCS of sand treated with PVA polymer after soaking in water for 3 min, as the concentration was increased from 0 to 3.7%. However, submerging sand treated with PMMA in water had no significant impact on the UCS. In addition, the UCS of three different types of soil (poorly graded sand, clayey sand with gravel, and sandy clay) treated with 0.002% of PAM and submerged in water of 25.4 mm in height on their sides for 30 min were reported by Georgees et al. (2017) [104] to decrease by approximately 50, 57 and 42% respectively. It was hypothesized that the polymer enwrapped the soil pores in such a way that water does not permeate into the inner pores. Thus, preventing the pore water pressure from developing and, as a result, increasing or maintaining the effective stress. The phenomenon that prevents water from penetrating the pores is connected to the increased viscosity of water when it encounters the polymer on the soil surface. As a result, water intrusion is reduced while the dry strength of the material is maintained. Naeini and Ghorbanalizadeh (2010) [183] reported a reduction in the UCS of sand treated with a combination of epoxy resin and polyamide hardener after immersion in water for 24 h. However, submerging the specimen beyond 24 h to 96 and 168 h increased the UCS. The increase in UCS was related to the interaction of the H+ ions of water in which the specimen was submerged with a three-member epoxies’ rings and the epoxies’ rings were opened, allowing the polyamide to readily react with the resin.

### 4.6. Long Term Stability

The curing process is one of the factors that influence the behavior of soil treated with polymers. A significant increase in the strength of soil treated with polymer with varying curing periods has been reported in several studies. Iyengar et al. (2013) [184] observed an improvement in the UCS of pavement subgrade treated with an acrylic-based polymer solution. Zandieh et al. (2010) [124] also noticed that the UCS of sandy soil improved significantly with an increase in the duration of curing of 1 to 28 days. In addition, similar observation has been reported for sand treated with PET [178], PU [100,161], PAM [105], and Styrene-Acrylate Copolymer SAC [105]. The UCS of sand treated with 10% of PU [100] and cured for a period of 2, 4, 7, and 28 days achieved about 95% of the total strength at 4 days. That of sand treated with PMMA and PVA [124] cured at 1, 3, 7, 14, and 28 days achieved about 87–96% of the total UCS at 7 days. In addition, PU-treated sand cured at 6, 12, 24, 48 and 72 h attained approximately 92–98% of the UCS at 48 h [161].

Also, the stability of fine-grain soil treated with polymer has been reported. According to Hasan and Shafiqu (2017) [143], the UCS of expansive clay treated with varying concentrations (6, 9, and 12%) of PE polymer at different curing times (0, 7, and 21 days) were observed to increase by 8–10%. However, the increase in strength with days is marginal. Several studies [97,103,108,110,126,139,142,147,153,154,155,156,158,159,166,185] has also reported increase in the UCS and shear strength parameter of polymer treated fine-grain soil as the curing period was increased. The curing process essentially comprised moisture evaporation from the specimens over time and hardening of the polymer-soil matrix. The reinforcing structure is not entirely developed during a shorter curing period. The moisture content reduces as curing time increases, and hence the strength rises. From the literature, it is evident that much polymer-treated soil can sustain its strength for a long period without deterioration.

Kou et al. (2020) [185], reported an increase in the permeability of marine clay treated with 2 g/kg of anionic PAM of approximately 19 times as the curing days increased from 7 to 28 days. At 7 days of curing time, the permeability of untreated marine clay is about 44 times greater than that of treated soil; however, at 14 and 28 days, the values are 5.5 and 2.3 times, respectively. The flocculation impact of the anionic PAM may have caused the initial decrease in permeability. However, as the number of curing days extended, the flocculation impact diminished, which may have led to a higher permeability of treated soil at a later stage of curing.

### 4.7. Erosion Resistance

Soil erosion is the removal and movement of soil particles caused by water runoff and wind forces. It poses a serious soil degradation danger to land, freshwater, and oceans.

#### 4.7.1. Water Erosion Resistance

Erosion induced by flowing water is a physical process that occurs as a result of the soil’s response to hydrodynamic stress, which is determined by the kinetics of the fluid that is often turbulent at the interface of a cohesive soil [186]. The removal of topsoil particles by water is more extensive and has a larger impact than wind removal [187]. Polymer addition to sand has been reported to demonstrate a strong erosion resistance against continuous runoff simulations compared to natural sand. The erosion of natural sand by the action of running water has been reported by Liu et al. (2019) [160] to take place immediately while PU polymer-treated sand showed a lower erosion ratio close to zero in the early erosion period. This indicates the delay of erosional behavior by polymer treatment. Furthermore, following the application of 5% PU polymer to the sand, the maximum erosion ratio per minute was reduced by 1.5 percent, and the occurrence time of erosion increased with PU concentration.

Similarly, Georgees et al. (2017) [104] also reported a significant reduction in soil mass loss following the treatment of three sandy soil with water-soluble anionic PAM. In other studies, Liu et al. (2011) [166] and Song et al. (2019) [165] reported a reduction in the erosion rate of clayey soil with an increase in the concentration of AEE and VAE polymer respectively. After the application of the stabilizer to the top of the clayey soil, the hydrophilic groups in the stabilizer are attached to the soil particles by hydrogen bonding and cation exchange. The hydrophobic C-C long chains group generates reticular membrane structures in the soil, gradually increasing the erosion resistance of the soil surface against induced runoff. Orts et al. (2007) [188] discovered that adding 10 ppm of anionic, high-quality PAM to irrigation water reduces sediment in runoff water by more than 90%. Furthermore, the application of the PAM polymer at construction sites and road cuts at a rate of 22.5 kg/ha^2^ resulted in a 60–85% reduction in sediment runoff during severe simulated rains [188]. Sojka et al. (1998) [189] also made a similar report on the effectiveness of PAM in reducing sediment runoff in water significantly.

#### 4.7.2. Wind Erosion Resistance

Wind erosion causes serious environmental and air-quality issues, garnering the attention of various researchers worldwide. It is a common occurrence in arid and semi-arid regions with flat, barren landscapes with loose, finely pulverized dry sandy soils [190]. When the minimal velocity necessary to displace soil particles is attained during wind erosion, soil particles disassociate from the soil mass. Polymers have also proven to be efficient to mitigate the effect of induced wind force on both sandy and clayey soil by generating polymer chains that produce a layer on the soil surface that contains aggregated soil particles. Bakhshi et al. (2021) [191] and Movahedan et al. (2012) [120] reported a reduction of more than 90% in erosion rates of PVA polymer-treated soil which proves the polymer efficiency in mitigating wind-induced erosion compared to those treated with water. The addition of PVA polymer [192] to sandy and loamy sandy soil significantly reduced the soil mass loss.

According to Movahedan et al. (2012) [120], the surface layer of polymeric samples of sandy soil after drying is entirely homogeneous, moderately rigid, and free of cracks. However, silty loam and silty clayey loam treated with polymer have a layer with irregular fissures. The fracture patterns in the water and polymer-treated samples are identical. Ding et al. (2020) [106] reported that the application of cationic PAM greatly reduced sand weight loss with higher concentration providing a better dust control performance with a longer protective period. Orts et al. (2007) [188] presented a 6:1:1 superabsorbent formulation of PAM combined with aluminum chlorohydrate and cross-linked poly(acrylic acid) superabsorbent to construct helicopter landing pads that decrease dust clouds during helicopter operation in fine, arid soils. Cationic PAM polymers with higher solution viscosity result in better crust strength and dust erosion resistance when applied on the red sand surface [107]. Han et al. (2007) [193] established in the field that PVAO and PVA polymer stabilizers could create a binding crust quickly and that the binding crusts were robust enough to stabilize the sand.

## 5. Reinforcement Mechanism of Polymer-Soil Composite

### 5.1. Cohesionless Soil

Sand has weak cohesion, a loose structure, and less clay content in general, but cohesion in polymer-treated sandy soil relies on membrane strength and the quality of the membrane enwrapping sand particles [182]. The different stages of the reinforcement mechanism of polymer in sand include void filling, chemical reaction, and enwrapping [117,118,160]. When the diluted polymer solution is mixed with sand and compacted, a part of them fills up the voids and others adsorb on the surface of the sand particle. Thus, a network of membranes is formed as the admixture loses moisture. This membrane structure increases the bond and the interlocking force between the sand particles which binds the loose sand particles together. Soil specimens treated with a higher polymer concentration have a more stable bridge with a broad adhesion surface contact via the soil particles and stronger membranes, which improves the strength performances [182]. During the amendment process of polymer-soil admixture, the physicochemical interaction between polymer solution and sand takes a while to be completed. The physicochemical response is intense during the first several hours, then weakens. As a result, the strength of reinforced sand increases considerably with curing time during the first 48 h, then gradually increases before reaching a constant value. The mechanical properties of sand such as compressive strength, tensile strength, permeability, erosion resistance, and cohesion may improve as a result of this void filling and physicochemical bonding [118].

### 5.2. Cohesive Soil

Similarly, the reinforcement mechanism of an organic polymer in clayey soil involves pore filling, chemical reaction, and enwrapping [121,157,166]. However, the chemical interaction of a polymer with clay particles is quite different compared to sand. The addition of polymer to clay alters the microstructural fabrics owing to induced nanocomposites [65]. According to Soltani et al. (2018) [102], the extent of polymer adsorption/attraction to clay particles is highly influenced by the type of polymer charge (cationic, non-ionic, or anionic). Cationic polymers (polycations) are electrostatically drawn to the negatively charged clay surface, whereas anionic polymers (polyanions) suffer less adsorption due to initial charge repulsion between the polymer and the negatively charged clay surface [194]. However, the presence of polyvalent cations can promote polyanion adsorption. Also, anionic polymer-clay interaction may be dominated by electrostatic attraction of the positively charged edge of clay surfaces [119]. Non-ionic polymers adsorb via van der Waals (dispersion) forces and/or hydrogen bonding [102,195]. When polymer solution is applied to clayey soil, the hydrophilic/hydrophobic groups in its molecular structure react with the ions of clay grain and create physicochemical bonds via ionic, electrostatic, cation bridge, ion-dipole interactions, hydrogen, or Van der Waals bonds [108,121,157,166,196]. Kang et al. (2016) [197] reported induced large face-to-face (FF) aggregations of kaolinite soil treated with PEO. The major interaction between the polymer and smectite, according to Deng et al. (2006) [198], is (i) ion-dipole interaction between exchangeable cations and the carbonyl (C=O) oxygens of amide groups (CONH2), which is likely more relevant for the transition-metal cation exchanged smectite, and (ii) hydrogen-bonding between the amide groups and water molecules in the hydration shells of exchangeable cations. Partially acetylated PVAC has been reported to be strongly and irreversibly adsorbed by montmorillonite via hydrogen bonds between the hydroxyl groups of the polymer and oxygens present on the clay surface [199].

## 6. Conclusions

The onus of the geotechnical engineer has been centered on economic and environmentally friendly alternatives to improve the physical properties of weak soil. Over the years, a few studies have been conducted to assess the feasibility of utilizing polymers to modify soil behavior. This review article summarizes details of synthetic polymers that have been employed to improve the geotechnical properties of soil by focusing on the durability aspects and the principle behind the reinforcement mechanism. The polymer inclusion within the soil matrix creates a network of membranes that acts as a binder creating a linkage between adjacent soil particles. In clay structure, the polymer tends to form a new composite material that alters the behavior of the soil. The following noteworthy conclusions are drawn from the review:The most widely used synthetic polymers in most geotechnical engineering application include PAM, PE, PU, PP, PVA, VC, AP, CBR plus, RPP, SS 299 and canlite.The soil-polymer interaction is dependent on the type of polymer, concentration, molecular weight, ionic charge, soil type, moisture content, curing period, water affinity, etc.The inclusion of a polymer in fine-grained soil can either increase, decrease, or have no significant impact on the Atterberg’s limits. Highly plastic clay when amended with PP polymer, can significantly reduce the Atterberg’s limits.Polymers play a significant role in altering the compaction characteristic of fine-grained soil following the formation of hydrophobic nanocomposite materials within the soil particles, which further act as nano-fillers. Application of VC and PP to fine-grained soil is effective in reducing the OMC and in increasing the MDD.The UCS of polymer-treated sand is found to improve significantly with polymer concentration. The increase in the strength has been linked to the formation of a polymer network membrane within the soil particles which improves the cohesive force between the soil particles resisting the shearing. The application of PE canlite, and VAE polymer to clayey soil; Probase, SS 299, and Canlite to silty soil; and SAE, PVA, PMMA, and SBR to sandy soil efficiently improves the UCS.The formation of a polymer network within the pore of a soil limits the seepage of water through the pores of the soil. The application of anionic PAM, VC, and PP to fine-grain soil and PAM, AP, SA, VA, and PU to sandy soil significantly reduces the hydraulic conductivity. Contrarily, VC polymer are found to increase the hydraulic conductivity of expansive clay.The reinforcement mechanism of both coarse and fine-grained soil treated with polymer includes pore filling, chemical reaction, and enwrapping. However, the chemical reaction of polymers with clay has been reported to be quite different from the case with coarse-grained soil. Cation exchange, electrostatic attraction, hydrogen bonding, van der Waals forces, and ion-dipole interactions are some of the mechanisms by which polymer molecules interact chemically with clay particle surfaces.The application of polymers to expansive clayey soil improves the volume stability and swell characteristics. Polymer such as POE, PAM, and PVA when amended with expansive clay, reduce the volumetric swell ratio (VSR), free swell ratio (FSR), and free swell index (FSI). Application of PP, PE, and PPMA to expansive clayey soil reduces the swell potential and pressure considerably.Certain polymers when treated with soil exhibit degradation in strength and stability when exposed to freeze- thaw, wet-dry cycles, and soaking because of their ability to absorb water, which results in the breakdown of polymer chain and resulting in the loss of strength. However, the application of PVA, VAEC, and PET polymer to sandy soil increases the durability against freeze-thaw action.The use of PVA and PMMA polymers to sandy soil enhances the thermal degradation. ADNB, AEE, crosslinked PAM-CMC, and PU are shown to exhibit excellent water stability under changing environments.PU, PAM, AEE, and VAE are effective in minimizing the water erosion, while PVA, PAM and PVAO have successfully been applied to significantly reduce soil mass loss.

## Figures and Tables

**Figure 3 polymers-14-05004-f003:**
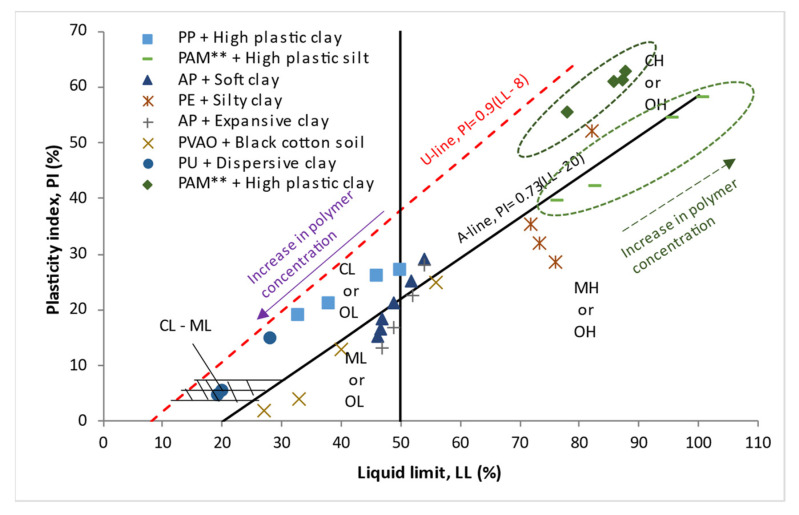
Plasticity chart of soil treated with polymer (** indicates polymer concentration as a % of weight of water). Note: this figure was created with data obtained from [65,102,108,141,142,143,145,147].

**Figure 4 polymers-14-05004-f004:**
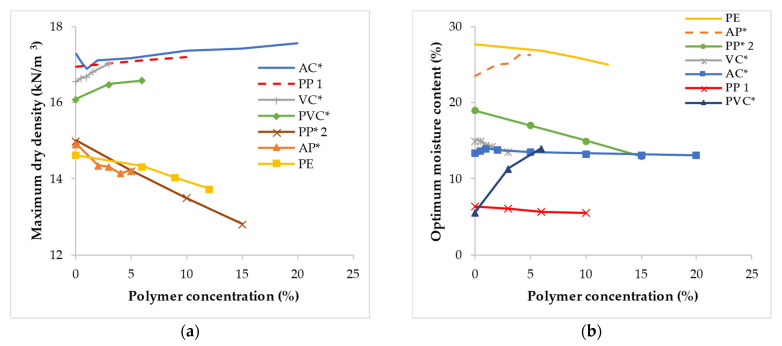
Variation in the compaction characteristic of fine-grain soil treated with polymer: (**a**) Maximum dry density; (**b**) Optimum moisture content (* indicates polymer concentration as a % of dry weight of soil). Note: these figures were created with data obtained from [65,109,110,136,143,149,150].

**Figure 6 polymers-14-05004-f006:**
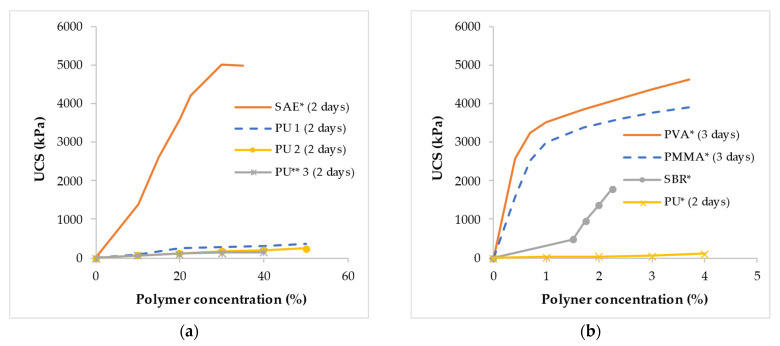
Influence of polymer treatment on the UCS of sand applied at: (**a**) High polymer concentration; (**b**) Low polymer concentration (* indicates polymer concentration as a % of dry weight of soil and ** as a % of weight of water). Note: these figures were created with data obtained from [114,116,124,128,130,160,161].

**Figure 7 polymers-14-05004-f007:**
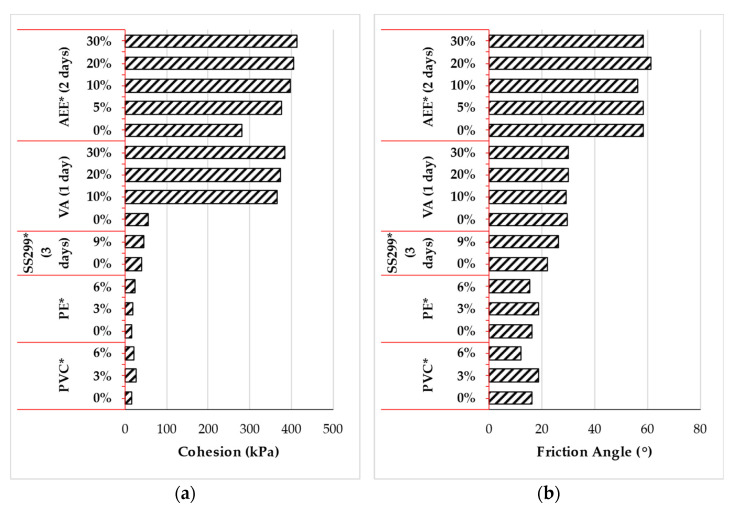
Variation in the shear strength parameter of fine-grain soil with polymer concentration: (**a**) Cohesion; (**b**) Frictional angle (* indicates polymer concentration as a % of dry weight of soil). Note: these figures were created with data obtained from [109,159,165,166].

**Figure 8 polymers-14-05004-f008:**
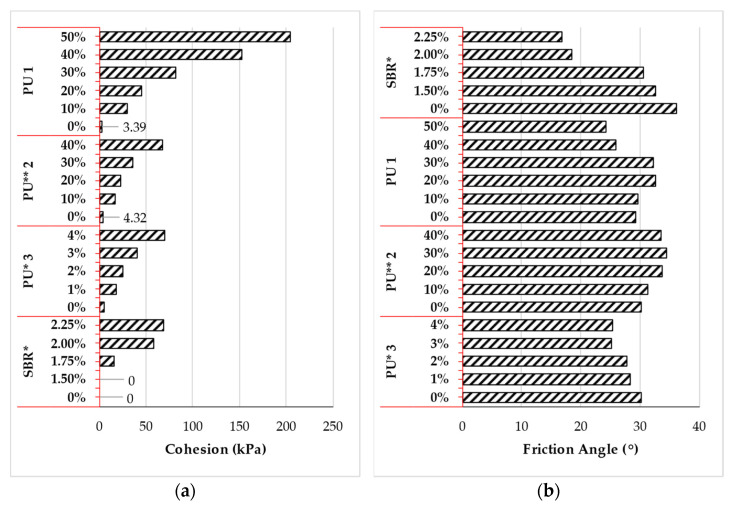
Variation in the shear strength parameter of cohesionless soil with polymer concentration: (**a**) Cohesion; (**b**) Frictional angle (* indicates polymer concentration as a % of dry weight of soil and ** as a % of weight of water). Note: these figures were created with data obtained from [114,116,130,160].

**Figure 9 polymers-14-05004-f009:**
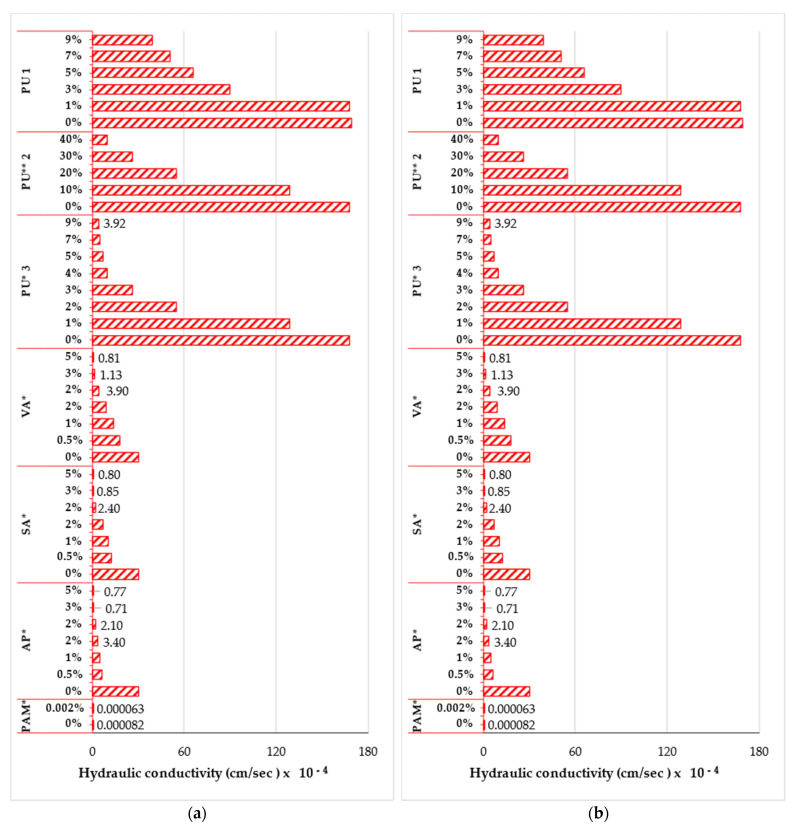
Variation of hydraulic conductivity with polymer concentration for: (**a**) Fine-grain soil; (**b**) Cohesionless soil (* indicates polymer concentration as a % of dry weight of soil and ** as a % of weight of water). Note: these figures were created with data obtained from [65,104,118,135,160,162,169,170,171,172].

**Figure 10 polymers-14-05004-f010:**
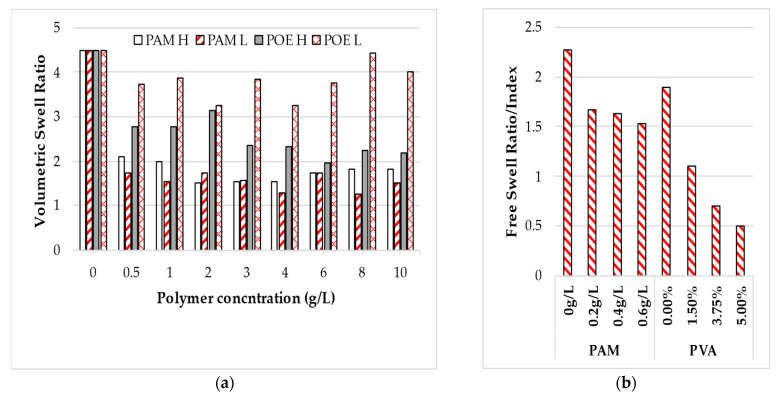
Variation in the sediment behavior of expansive clay with polymer concentration: (**a**) Volumetric swell ratio; (**b**) Free swell ratio/index (concentration of PVA as a % of dry weight of soil). Note: these figures were created with data obtained from [101,102,122].

**Figure 11 polymers-14-05004-f011:**
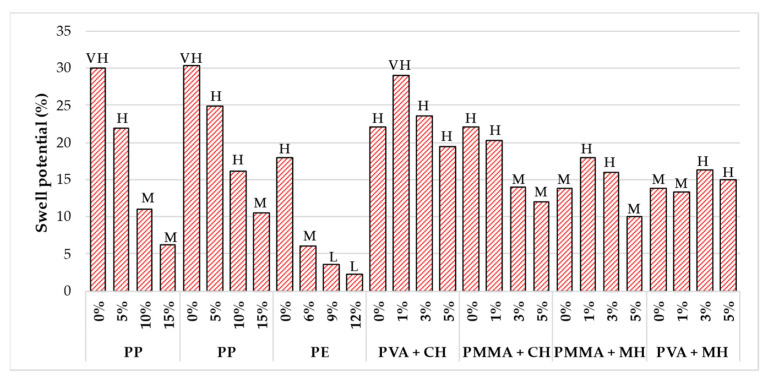
Influence on polymer treatment on the swell potential of expansive clay (concentration of PP, PVA and PMMA are defined as the % of the dry weight of soil). Note: this figure was created with data obtained from [110,143,171,177].

**Figure 12 polymers-14-05004-f012:**
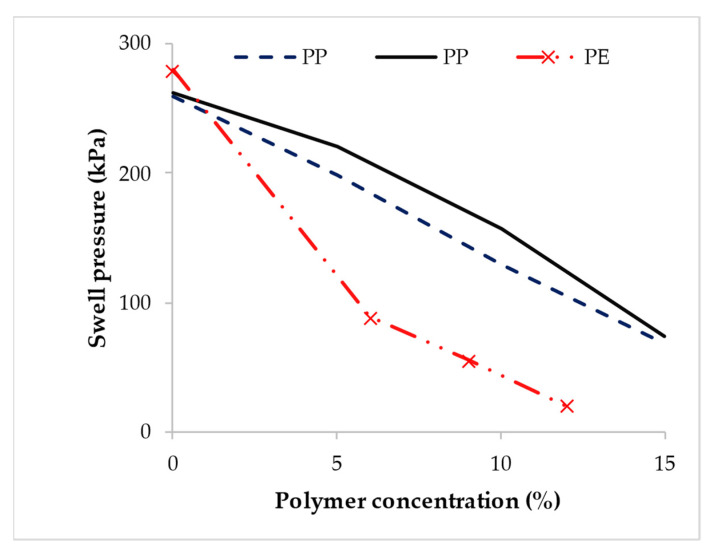
Influence on polymer treatment on the swell pressure of expansive clay (concentration of PP is defined as the % of the dry weight of soil). Note: this figure was created with data obtained from [110,143,171].

**Table 2 polymers-14-05004-t002:** The effect of various polymer on CBR.

Reference	Soil	Polymer Type	Polymer Content (%)	Penetration (mm)	CBR (%)
[95]	High plastic clay	CBR Plus	0, 0.0096, 0.03, & 0.05. (% by weight of water)	2.54	13, 22.4, 24.5, & 27.5
		RPP	0, 0.019, 0.04, & 0.06 (% by weight of water)		18, 21.7, 24.7, & 25.5
[105]	Silty sand	Anionic PAM	0, 0.001, 0.002, & 0.003 (% by weight of dry soil)	-	19.05, 24.39, 25.03, & 23.31 (unsoaked)
					19.08, 18.35, 25.25, & 24.24 (Soaked)
		SAC	0, 0.5, 0.7, & 1 (% by weight of dry soil)		19.05, 23.88, 20.11, & 11.11 (unsoaked)
					19.08, 19.53, 18.35, & 9.68 (Soaked)
[136]	High plastic clay	VC	0, 0.5, 1, 1.5, & 3 (% by weight of dry soil)	2.54	6.50, 11.40, 12.10, 14.70, & 12.50 (Unsoaked)
				5.08	7.13, 12.47, 13.07, 16.40, & 13.47 (Unsoaked)
[143]	High plastic clay	HDPE	0, 6, 9, & 12	-	5.5, 9.1, 15, & 21.5
[131]	Sand	UFR	0, 1, &2 (% by weight of dry soil)	-	12.25, 20.10, & 25.35
		SBR	0, 1, 2 & 3 (% by weight of dry soil)		12.25, 6.59, 11.69, & 14.39

**Table 3 polymers-14-05004-t003:** Classification procedures for expansive soils. Adapted with permission Sridharan and Prakash (2000) [138] © ICE Publishing (2000).

Oedometer Swell (%)	Soil Expansivity
<1	Negligible
1–5	Low (L)
5–15	Moderate (M)
15–25	High (H)
>25	Very high (VH)

**Table 4 polymers-14-05004-t004:** Effect of different polymers on the compression index of soil.

Reference	Soil	Polymer Type	Polymer Content (%)	Compression Index, C_c_
[109]	High plastic clay	PVC	0, 3, & 6 (% by weight of dry soil)	0.33, 0.19, & 0.25
		HDPE		0.33, 0.22, & 0.40
[111]	Low plastic clay	PP	0, 1.5, 3, & 5 (% by weight of dry soil)	0.19, 0.14, 0.13, & 0.12
[119]	Kaolin	MBA	0, & 5% (% by weight of water)	0.19, & 0.17
		STBA		0.19, & 0.13
		PVA		0.19, & 0.174

## Data Availability

Not applicable.

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
