# Peer review of "A Critical Review on the Feasibility of Synthetic Polymers Inclusion in Enhancing the Geotechnical Behavior of Soils"

_polymers, 2022, doi:10.3390/polym14225004_

Round 1

Reviewer 1 Report

The manuscript is a comprehensive summary of synthetic polymer for ground improvement in geotechnical applications. The comments are as follows:

General comment:

1.      Categorize and simplify the title and subtitle, based on general geotechnical background, to unify the content of critical review.

2.      Many definitions of parameters considered are missed and not clear.

3.      Add the reference of the reported data presented in all figures.

4.      Table 1 summarizes the geotechnical properties commonly studied in literatures. Specific examples are permeability (water flow in soil), compression index (soil stiffness), CBR (soil strength), which are key parameters to estimate the improvement of stabilized soils in geotechnical engineering. Add the summary of the three parameters by using figures or tables.

Specific comments:

1.      Currently classification of polymers/materials for the chemical stabilizations of soils is not clear. Precisely classify in terms of 1) conventional and unconventional polymers, 2) natural and synthetic polymers, 3) new and recycled materials in each paragraph.

2.      In the beginning part of section 2, the description of “synthetic polymer” should be given, instead of general “polymer”. Also, the order of polymer types should coincide with the subsection titles (2.1-2.8).

3.      Provide the numerical level of high and low concentrations of polymers presented in several figures (e.g. figures 1, 2 and 4). Also, clearly explain the “definition” of concentration including volume or weight basis.

4.      Sections 3.4 and 3.5 should be combined, where volume change (swelling) is considered. Also, there are many parameters related to soil swelling, Accordingly, the definition of swelling parameters mentioned in this study should be presented.

Author Response

Response to Reviewer #1 Comments

The authors are grateful to the anonymous reviewers for their respective comments to improve the quality of this review paper. Certain figures are now modified/edited in this revised version of manuscript. The authors have addressed all the comments raised, and the amendments are highlighted in red in the revised manuscript

General Comment

  1. Categorize and simplify the title and subtitle, based on general geotechnical background, to unify the content of critical review.

Response: Agreed. The title has now been modified to “A Critical Review on the Feasibility of Synthetic Polymers Inclusion in Enhancing the Geotechnical Behavior of Soils”.

  1. Many definitions of parameters considered are missed and not clear.

Response:  Clarification. The definitions of the parameters are duly provided in the first instance they appear in the manuscript.

  1. Add the reference of the reported data presented in all figures.

Response: Agreed and Revised as Suggested. The authors appreciate the reviewer for this comment and the references have now been added relevant figures in the revised manuscript.

  1. Table 1 summarizes the geotechnical properties commonly studied in literatures. Specific examples are permeability (water flow in soil), compression index (soil stiffness), CBR (soil strength), which are key parameters to estimate the improvement of stabilized soils in geotechnical engineering. Add the summary of the three parameters by using figures or tables.

Response: The authors are grateful to the reviewer for this suggestion. A new section (3.3.3) has been included for the California Bearing Ration (CBR) and Table 2 has also been included, which summarizes the result of CBR from the literature. The authors would like to bring to the kind attention of the reviewer, the fact that, a section summarizing the results of the influence of synthetic polymer on the hydraulic conductivity has already been included in section 3.4 in the first submitted version. Lastly, a section summarizing the compression index has been included in section 3.5.3 as shown in line no 791 to 803. The summary of the result has been presented in Table 4.

Specific Comments

  1. Currently classification of polymers/materials for the chemical stabilizations of soils is not clear. Precisely classify in terms of 1) conventional and unconventional polymers, 2) natural and synthetic polymers, 3) new and recycled materials in each paragraph.

Response: The authors thank the reviewer for this suggestion. However, in line 41-57, the chemical stabilizers have been classified based on conventional and unconventional stabilizers. Also, in line 80 – 89, the classification of polymers into natural and synthetic polymers has been duly mentioned. Further, elaboration on the new and recycled materials used in geotechnical engineering is beyond the scope of this review paper and accordingly it has not been included.

  1. In the beginning part of section 2, the description of “synthetic polymer” should be given, instead of general “polymer”. Also, the order of polymer types should coincide with the subsection titles (2.1-2.8).

Response: Agreed and Revised as Suggested. Necessary correction has been done as seen in line 102-110.

  1. Provide the numerical level of high and low concentrations of polymers presented in several figures (e.g., figures 1, 2 and 4). Also, clearly explain the “definition” of concentration including volume or weight basis.

Response: Agreed and Included in the revised version of this manuscript. The numerical level of high and low concentrations of polymers presented in figures 1, and 2 are now included in the context of the revised manuscript in lines 294 – 295. However, Figure 4 does not present a high and low concentrations of polymers. Probably the reviewer was referring to Figure 6, and necessary corrections has been done and shown in lines 488 and 489 respectively.

  1. Sections 3.4 and 3.5 should be combined, where volume change (swelling) is considered. Also, there are many parameters related to soil swelling, Accordingly, the definition of swelling parameters mentioned in this study should be presented.

Response: The authors thank the reviewer for this constructive comment. The author agrees completely with the reviewer and necessary corrections have been made. It is assumed that the reviewer meant section 3.5 (Sediment volume behavior) and 3,6 (Swell characteristics). Thus, section 3.5 has been divided into two sub-sections (3.51 and 3.52). See line 655 and 689, respectively.

Reviewer 2 Report

The work is well-thought-out and written in an accessible language. Provides a collection of the most important information in the field of synthetic polymers for soil improvement. The authors cite 7 of their own works, which with the total number of citations (195) is adequate. The work is extremely extensive, but it is easy to read. It is difficult to find any major reservations (among others due to the review nature of the work). The article has editorial shortcomings, due to the length of the article, I will allow myself not to mention them. I only ask the Authors to carefully edit the article.

Author Response

Response to Reviewer # 2 Comments

The authors are grateful to the anonymous reviewers for their respective comments to improve the quality of this review paper. Certain figures are now modified/edited in this revised version of manuscript. The authors have addressed all the comments raised, and the amendments are highlighted in red in the revised manuscript

Comments and Suggestions for Authors

The work is well-thought-out and written in an accessible language. Provides a collection of the most important information in the field of synthetic polymers for soil improvement. The authors cite 7 of their own works, which with the total number of citations (195) is adequate. The work is extremely extensive, but it is easy to read. It is difficult to find any major reservations (among others due to the review nature of the work). The article has editorial shortcomings, due to the length of the article, I will allow myself not to mention them. I only ask the Authors to carefully edit the article.

Response: The authors are extremely grateful to the reviewer for his constructive critique and valuable insight. As suggested, the entire manuscript has now been proofread for consistency in English, punctuation and grammar. All the references are duly checked and tagged to respective figures in this revised version. The constructive comments have raised the morale of the authors while revising this manuscript. 

Reviewer 3 Report

The chemical and biological effects of  these polymers were not evaluated and for this reason  the environmental friendly factor of these materials are not clear. On the other hand, you reviewed only physical parameters  while  chemical effects of  synthetic polymers more important for soil biomass and should be evaluate.Some physical parameters such as COLE coefficient and soil aggregate stability did not show. Many figures did not have any references.

Author Response

Response to Reviewer #3 Comments

The authors are grateful to the anonymous reviewers for their respective comments to improve the quality of this review paper. Certain figures are now modified/edited in this revised version of manuscript. The authors have addressed all the comments raised, and the amendments are highlighted in red in the revised manuscript

Comments and Suggestions for Authors

The chemical and biological effects of these polymers were not evaluated and for this reason the environmental friendly factor of these materials are not clear. On the other hand, you reviewed only physical parameters while chemical effects of synthetic polymers more important for soil biomass and should be evaluate. Some physical parameters such as COLE coefficient and soil aggregate stability did not show. Many figures did not have any references.

Response:  The authors thank the reviewer for his insight and suggestion. The authors wish to provide this clarification.

  • The biological effects are of importance for biodegradable polymers like chitosan, casein, xanthan gum and guar gum. The current study is limited to only synthetic polymers which are known to withstand the long-term biodegradable effects of water and wind erosion.
  • Secondly, this review is intended and aimed to address only the enhancement in geotechnical properties of synthetic polymer amended soils and accordingly the relevant physical properties are critically discussed and evaluated in the revised version of this manuscript.
  • COLE (Coefficient of Linear Extensibility) is an inherent function of minerals present in the soil. The addition of synthetic polymers does not alter or modify the natural minerals present in any given soil unlike pozzolanic stabilization involving lime, cement or flyash. The modification in geotechnical properties with the addition of synthetic polymers is due to the formation of polymeric substance which coats the surface of soil particles and does not interfere with the natural minerals in any form. Accordingly, this aspect has not been discussed in this review article as it is not directly applicable.
  • The soil aggregate stability has been duly discussed in sub-sections 4.4, 4.5, 4.6 and 4.7.1 of the revised version of this manuscript.
  • The references for all the figures are now duly mapped in this revised version of manuscript.

Round 2

Reviewer 1 Report

The reviewer has examined the revision made by the authors and confirmed that most comments are reflected in the revised manuscript. However, the definition of the concentration is not still clear. The concentration can be defined in terms of “polymer/(polymer+soil+water)” or “polymer/soil” and so on. Also, the ratio is expressed in terms of “volume” or “weight/mass”. The behaviors of soils modified with polymer are highly dependent on how the concentration defines. Moreover, the concentration of all existing data used in this study should be converted to be equivalent to the concentration the authors defines and reflected in the text and figures. This is because each previous studies define the concentration in its own way.

Author Response

Response to Reviewer #1 Comments (Round 2)

The authors are grateful to the anonymous reviewers for their respective comments to improve the quality of this review paper. The authors have addressed all the comments raised and comprehensively revised the manuscript. Certain figures are now modified/edited in this revised version of manuscript. The annotated version of manuscript is provided by using “RED” color font.

Query: The reviewer has examined the revision made by the authors and confirmed that most comments are reflected in the revised manuscript. However, the definition of the concentration is not still clear. The concentration can be defined in terms of “polymer/(polymer+soil+water)” or “polymer/soil” and so on. Also, the ratio is expressed in terms of “volume” or “weight/mass”. The behaviors of soils modified with polymer are highly dependent on how the concentration defines. Moreover, the concentration of all existing data used in this study should be converted to be equivalent to the concentration the authors defines and reflected in the text and figures. This is because each previous studies define the concentration in its own way.

Response: The authors are thankful to the reviewer for raising this important point and would like to state the following:

  • The polymer concentration applied in most cases in the journal papers reviewed were presented as a percentage by the weight of the dry soil and in few cases as a percentage of the weight of water used to prepare the polymer. However, in some cases the concentration of the polymer was not clearly defined by the respective authors.
  • The authors would also like to inform the reviewer that in the cases where the polymer concentration was defined as g/L or mg/L in the papers reviewed, it has been converted to equivalent concentration as the percentage by weight of water applied while preparing this revised version of manuscript.
  • Finally, the figures in the manuscript have been modified to indicate the precise definition of each polymer(s) presented. The polymers that were not clearly defined (by the authors of those papers) have been left as they are in the previous version of the manuscript.

Reviewer 3 Report

The comparison between dry and wet sieving of soil samples with polymers and shaking them in water and alcohol will show that which soil polymers will be better.

Author Response

Response to Reviewer #3 Comments (Round 2)

The authors are grateful to the anonymous reviewers for their respective comments to improve the quality of this review paper. The authors have addressed all the comments raised and comprehensively revised the manuscript. Certain figures are now modified/edited in this revised version of manuscript. The annotated version of manuscript is provided by using “RED” color font.

Query: The comparison between dry and wet sieving of soil samples with polymers and shaking them in water and alcohol will show that which soil polymers will be better.

Response:  The authors appreciate the reviewer for his/her comment. However, the authors would like to bring to the kind notice of the reviewer the fact that, the current review paper (second revised) does not mention in any section, an attempt to identify the type of polymer present in the soil via the process of dry (or) wet sieving. The authors have critically gone through all the papers cited in the second revised version of manuscript and didn’t find a single instance where such an exercise has been carried out by the respective authors of the manuscript.

Round 3

Reviewer 1 Report

The revision is made perperly by reflecting the 2nd comments.